# TRUNCATED CONSISTENCY MODELS

**Sangyun Lee**[*]
Carnegie Mellon University

**Yilun Xu**
NVIDIA

**Tomas Geffner**
NVIDIA

**Giulia Fanti**
Carnegie Mellon University

**Karsten Kreis**
NVIDIA

**Arash Vahdat**
NVIDIA

**Weili Nie**
NVIDIA

## ABSTRACT

Consistency models have recently been introduced to accelerate sampling from diffusion models by directly predicting the solution (i.e., data) of the probability flow ODE (PF ODE) from initial noise. However, the training of consistency models requires learning to map all intermediate points along PF ODE trajectories to their corresponding endpoints. This task is much more challenging than the ultimate objective of one-step generation, which only concerns the PF ODE's noise-to-data mapping. We empirically find that this training paradigm limits the one-step generation performance of consistency models. To address this issue, we generalize consistency training to the truncated time range, which allows the model to ignore denoising tasks at earlier time steps and focus its capacity on generation. We propose a new parameterization of the consistency function and a two-stage training procedure that prevents the truncated-time training from collapsing to a trivial solution. Experiments on CIFAR-10 and ImageNet $64 \times 64$ datasets show that our method achieves better one-step and two-step FIDs than the state-of-the-art consistency models such as iCT-deep, using more than $2\times$ smaller networks. Project page: https://truncated-cm.github.io/

## 1 INTRODUCTION

Diffusion models (Ho et al., 2020; Song et al., 2020) have demonstrated remarkable capabilities in generating high-quality continuous data such as images, videos, or audio (Ramesh et al., 2022; Ho et al., 2022; Huang et al., 2023). Their generation process gradually transforms a simple Gaussian prior into data distribution through a probability flow ordinary differential equation (PF ODE). Although diffusion models can capture complex data distributions, they require longer generation time due to the iterative nature of solving the PF ODE.

Consistency models (Song et al., 2023) were recently proposed to expedite the generation speed of diffusion models by learning to directly predict the solution of the PF ODE from the initial noise in a single step. To circumvent the need for simulating a large number of noise-data pairs to learn this mapping, as employed in prior works (Liu et al., 2022b; Luhman & Luhman, 2021), consistency models learn to minimize the discrepancy between the model's outputs at two neighboring points along the ODE trajectory. The boundary condition at $t = 0$ serves as an anchor, grounding these outputs to the real data. Through simulation-free training, the model gradually refines its mapping at different times, propagating the boundary condition from $t = 0$ to the initial $t = T$.

However, the advantage of simulation-free training comes with trade-offs. Consistency models must learn to map any point along the PF ODE trajectory to its corresponding data endpoint, as shown in Fig. 1a. This requires the learning of both *denoising* at smaller times on the PF ODE, where the data are only partially corrupted, and *generation* towards $t = T$, where most of the original data information has been erased. This dual task necessitates larger network capacity, and it is challenging for a single model to excel at both tasks. Our empirical observations in Fig. 2 demonstrate the model would gradually sacrifice its denoising capability at smaller times to trade for generation quality as training proceeds. While this behavior is desirable as the end goal is generation rather than denoising, we argue for explicit control over this trade-off, rather than allowing the model to allocate capacity

---

[*]Work mostly done while interning at NVIDIA

uncontrollably across times. This raises a key question: *Can we explicitly reduce the network capacity dedicated to the denoising task in order to improve generation?*

In this paper, we propose a new training algorithm, termed ***Truncated Consistency Models*** (TCM), to de-emphasize denoising at smaller times while still preserving the consistency mapping for larger times. TCM relaxes the original consistency objective, which requires learning across the entire time range $[0, T]$ of PF ODE trajectories, to a new objective that focuses on a truncated time range $[t', T]$, where $t'$ serves as the dividing time between denoising and generation tasks. This allows the model to dedicate its capacity primarily to generation, freeing it from the denoising task at earlier times $[0, t')$. Crucially, we show that a proper boundary condition at $t'$ is necessary to ensure the new model adheres to the original consistent mapping. To achieve this, we propose a two-stage training procedure (see Fig. 1a): The first stage involves pretraining a standard consistency model over the whole time range. This pretrained model then acts as the boundary condition at $t'$ for the subsequent truncated consistency training stage of the TCM.

Experimentally, TCM improves both the sample quality and the training stability of consistency models across different datasets and sampling steps. On CIFAR-10 and ImageNet $64 \times 64$ datasets, TCM outperforms the iCT (Song & Dhariwal, 2023), the previous best consistency model, in both one-step and two-step generation using similar network size. TCM even outperforms iCT-deep that uses a $2\times$ larger network across datasets and sampling steps. By using our largest network, we achieve a one-step FID of 2.20 on ImageNet $64 \times 64$, which is competitive with the current state-of-the-art. In addition, the divergence observed during standard consistency training is not present in TCM. We show through extensive ablation experiments why the various design choices of truncated consistency models (including the strength of mandating boundary conditions, two-stage training, etc.) are necessary to obtain these results.

**Contributions.** *(i)* We identify an underlying trade-off between denoising and generation within consistency models, which negatively impacts both stability and generation performance. *(ii)* Building on these insights, we introduce Truncated Consistency Models, a novel two-stage training framework that explicitly allocates network capacity towards generation while preserving consistency mapping. *(iii)* Extensive validation of TCM demonstrates significant improvements in both one-step and two-step generation, achieving state-of-the-art results within the consistency models family on multiple image datasets. Additionally, TCM exhibit improved training stability. *(iv)* We provide in-depth analyses, along with ablation and design choices that demonstrate the unique advantages of the two-stage training in TCM.

## 2 PRELIMINARIES

### 2.1 DIFFUSION MODELS

Diffusion models are a class of generative models that synthesize data by reversing a forward process in which the data distribution $p_{\text{data}}$ is gradually transformed into a tractable Gaussian distribution. In this paper, we use the formulation proposed in Karras et al. (2022), where the forward process is defined by the following stochastic differential equation (SDE):

$$d\mathbf{x}_t = \sqrt{2t}d\mathbf{w}_t, \tag{1}$$

where $t \in [0, T]$ and $\mathbf{w}_t$ is the standard Brownian motion from $t = 0$ to $t = T$. Here, we define $p_t$ as the marginal distribution of $\mathbf{x}_t$ along the forward process, where $p_0 = p_{\text{data}}$. In this case, $p_t$ is a perturbed data distribution with the noise from $\mathcal{N}(0, t^2\mathbf{I})$. In diffusion models, $T$ is set to be large enough so that $p_T$ is approximately equal to a tractable Gaussian distribution $\mathcal{N}(0, T^2\mathbf{I})$.

Diffusion models come with the reverse probability flow ODE (PF ODE) that starts from $t = T$ to $t = 0$ and yields the same marginal distribution $p_t$ as the forward process in Eq. (1) (Song et al., 2020):

$$d\mathbf{x}_t = -t\mathbf{s}_t(\mathbf{x}_t)dt, \tag{2}$$

where $\mathbf{s}_t(\mathbf{x}_t) := \nabla_{\mathbf{x}} \log p_t(\mathbf{x})$ is the score function at time $t \in [0, T]$. To draw samples from the data distribution $p_{\text{data}}$, we first train a neural network to learn $\mathbf{s}_t(\mathbf{x})$ using the denoising score matching (Vincent, 2011), initialize $\mathbf{x}_T$ with a sample from $\mathcal{N}(0, T^2\mathbf{I})$, and solve the PF ODE backward in time: $\mathbf{x}_0 = \mathbf{x}_T + \int_T^0 (-t\mathbf{s}_t(\mathbf{x}_t))dt$. However, numerically solving the PF ODE requires multiple forward passes of the neural score function estimator, which is computationally expensive.

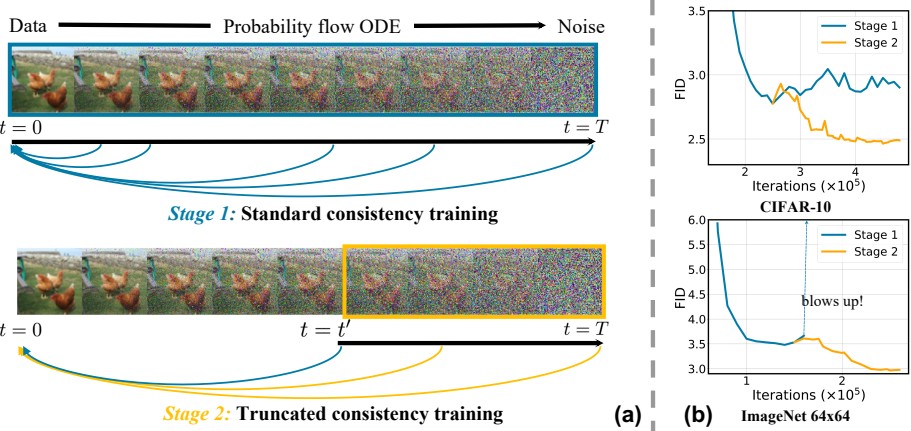

Figure 1: **(a)** Two-stage training of TCM. In Stage 1, a standard consistency model is trained to provide both the boundary condition and initialization for TCM training in Stage 2. TCM focuses on learning in the $[t', T]$ range, discarding denoising tasks at earlier times and allocating network capacity toward generation-like tasks at later times. **(b)** Sample quality (FID, lower is better) of the two training stages. TCM (Stage 2) improves over standard consistency training (Stage 1) across datasets. Additionally, standard consistency training shows instability on challenging datasets like ImageNet 64x64, where the model could diverge during training.

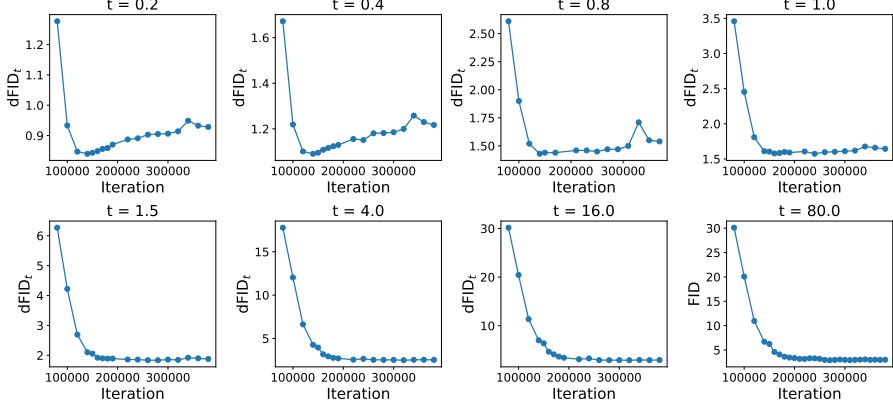

Figure 2: Evolution of the denoising FID ($\mathrm{dFID}_t$) during standard consistency training for different $t$, where $0 < t \leq 80$ follows the EDM noise schedule (Karras et al., 2022). The model gradually sacrifices its denoising capability at smaller times ($t < 1.0$) to trade for the improved generation quality at $t = 80$ as training proceeds.

## 2.2 CONSISTENCY MODELS

Consistency models instead aim to directly map from noise to data, by learning a consistency function that outputs the solution of PF ODE starting from any $t \in [0, T]$. The desired consistency function $\mathbf{f}$ should satisfy the following two properties (Song et al., 2023): (i) $\mathbf{f}(\mathbf{x}_0, 0) = \mathbf{x}_0$, and (ii) $\mathbf{f}(\mathbf{x}_t, t) = \mathbf{f}(\mathbf{x}_s, s)$, $\forall (s, t) \in [0, T]^2$. The first condition can be satisfied by the reparameterization

$$\mathbf{f}_{\boldsymbol{\theta}}(\mathbf{x}, t) := c_{\mathrm{out}}(t)\mathbf{F}_{\boldsymbol{\theta}}(\mathbf{x}, t) + c_{\mathrm{skip}}(t)\mathbf{x}, \qquad (3)$$

where $\boldsymbol{\theta}$ is the parameter of the free-form neural network $\mathbf{F}_{\boldsymbol{\theta}} : \mathbb{R}^d \times \mathbb{R} \to \mathbb{R}^d$, and $c_{\mathrm{out}}(0) = 0$, $c_{\mathrm{skip}}(0) = 1$ following the similar design of Karras et al. (2022). Here, instead of training $\mathbf{f}_{\boldsymbol{\theta}}$ directly, we train a surrogate neural network $\mathbf{F}_{\boldsymbol{\theta}}$ under the above reparameterization. The second condition can be learned by optimizing the following *consistency training* objective:

$$\mathcal{L}_{\mathrm{CT}}(\mathbf{f}_{\boldsymbol{\theta}}, \mathbf{f}_{\boldsymbol{\theta}}^-) := \mathbb{E}_{t \sim \psi_t, \mathbf{x} \sim p_{\mathrm{data}}, \boldsymbol{\epsilon} \sim \mathcal{N}(0, \mathbf{I})} \big[ \frac{\omega(t)}{\Delta_t} d(\mathbf{f}_{\boldsymbol{\theta}}(\mathbf{x} + t\boldsymbol{\epsilon}, t), \mathbf{f}_{\boldsymbol{\theta}^-}(\mathbf{x} + (t - \Delta_t)\boldsymbol{\epsilon}, t - \Delta_t)], \quad (4)$$

where $\boldsymbol{\theta}^- = \mathrm{stopgrad}(\boldsymbol{\theta})$, $\psi_t$ denotes the probability of sampling time $t$ that also represents the noise scale, $\boldsymbol{\epsilon}$ denotes the standard Gaussian noise, $\omega(t)$ is a weighting function, $d(\cdot, \cdot)$ is a distance

function defined in Sec. D.1.2, and $\Delta_t$ represents the nonnegative difference between two consecutive time steps that is usually set to a monotonically increasing function of $t$.

The gradient of $\mathcal{L}_{\text{CT}}$ with respect to $\boldsymbol{\theta}$ is an approximation of the underlying *consistency distillation* loss with a $O(\max_t \Delta_t)$ error (See Appendix D). Song et al. (2023) empirically suggests that $\Delta_t$ should be large at the beginning of training, which incurs biased gradients but allows for stable training, and should be annealed in the later stages, which reduces the error term but increases variance.

**Denoising FID**   By definition, consistency models can both generate data from pure Gaussian noise as well as noisy data sampled from $p_t$ where $0 < t < T$. To understand how consistency models propagate end solutions through diffusion time, we need to empirically measure their denoising capability across different time steps. To this end, we define denoising FID at time step $t$, termed $\text{dFID}_t$, as the Fréchet inception distance (FID) (Heusel et al., 2017) between the original data $p_{\text{data}}$ and the denoised data by consistency models with inputs sampled from $p_t$. When computing $\text{dFID}_t$, we first add Gaussian noise from $\mathcal{N}(0, t^2\mathbf{I})$ to 50K clean samples and then denoise them using consistency models. Hence, $\text{dFID}_0$ is close to zero, and $\text{dFID}_T$ is the standard FID.

## 3   TRUNCATED CONSISTENCY MODEL

Standard consistency models pose a higher challenge in training than many other generative models: instead of simply mapping noise to data, consistency models must learn the mapping from *any* point along the PF ODE trajectory to its data endpoint. Hence, a consistency model must divide its capacity between denoising tasks (i.e., mapping samples from intermediate times to data) and generation (i.e., mapping from pure noise to data). This challenge mainly contributes to consistency models' underperformance relative to other generative models with similar network capacities (see Table 1).

Interestingly, standard consistency models navigate the trade-off between denoising and generation tasks implicitly. We observe that during standard consistency training, the model gradually loses its denoising capabilities at the low $t$. Specifically, Fig. 2 shows a trade-off in which, after some training iterations, denoising FIDs at lower $t$ ($t < 1$) *increase* while the denoising FIDs at larger $t$ ($t > 1$) (including the generation FID at the largest $t = 80$) continue to decrease. This suggests that the model struggles to learn to denoise and generate simultaneously, and sacrifices one for the other.

Truncated consistency models (TCM) aim to explicitly control this tradeoff by forcing the consistency training to ignore the denoising task for small values of $t$, thus improving its capacity usage for generation. We thus *generalize* the consistency model objective in Eq. (4) and apply it only in the truncated time range $[t', T]$ where the dividing time $t'$ lies within $(0, T)$. The time probability $\psi_t$ in TCM only has support in $[t', T]$ as a result.

**Naive solution**   A straightforward approach is to directly train a consistency model on the truncated time range. However, the model outputs can collapse to an arbitrary constant because a constant function (i.e., $\mathbf{f}_{\boldsymbol{\theta}}(\mathbf{x}, t) = \text{const}$) is a minimizer of the consistency training objective (Eq. (4)). In standard consistency models, the boundary condition $\mathbf{f}(\mathbf{x}_0, 0) = \mathbf{x}_0$ prevents collapse, but in this naive example, there is no such meaningful boundary condition. For example, if the free-form neural network $\mathbf{F}_{\boldsymbol{\theta}}(\mathbf{x}, t) = -c_{\text{skip}}(t)\mathbf{x}/c_{\text{out}}(t)$ for all $t \in [t', T]$, $\mathbf{f}_{\boldsymbol{\theta}}(\mathbf{x}, t)$ is $\mathbf{0}$, and thus Eq. (4) becomes zero. To handle this, we propose a two-stage training procedure and design a new parameterization with a proper boundary condition, as outlined below.

**Proposed Solution**   Truncated consistency models conduct training in two stages:

1. **Stage 1 (Standard consistency training):** We pretrain a consistency model to convergence in the usual fashion, with the training objective in Eq. (4); we denote the pre-trained model as $\mathbf{f}_{\boldsymbol{\theta}_0}$.

2. **Stage 2 (Truncated consistency training):** We initialize a new consistency model $\mathbf{f}_{\boldsymbol{\theta}}$ with the first-stage pretrained weights $\mathbf{f}_{\boldsymbol{\theta}_0}$, and train over a truncated time range $[t', T]$. The boundary condition at time $t'$ is provided by the pretrained $\mathbf{f}_{\boldsymbol{\theta}_0}$. This stage is explained further below.

To explain the details of TCM, we first introduce the following parameterization:

$$\mathbf{f}_{\boldsymbol{\theta}, \boldsymbol{\theta}_0^-}^{\text{trunc}}(\mathbf{x}, t) = \mathbf{f}_{\boldsymbol{\theta}}(\mathbf{x}, t) \cdot \mathbb{1}\{t \geq t'\} + \mathbf{f}_{\boldsymbol{\theta}_0^-}(\mathbf{x}, t) \cdot \mathbb{1}\{t < t'\}, \tag{5}$$

where $\mathbb{1}\{\cdot\}$ is the indicator function, and similarly, $\boldsymbol{\theta}_0^- = \text{stopgrad}(\boldsymbol{\theta}_0)$. Intuitively, we only use our final model $\mathbf{f}_{\boldsymbol{\theta}}$ when $t \geq t'$, and we inquire the pre-trained $\mathbf{f}_{\boldsymbol{\theta}_0^-}$ otherwise. This approach ensures that (1) $\mathbf{f}_{\boldsymbol{\theta}}$ does not waste its capacity learning in the $[0, t')$ range, and (2) if $\mathbf{f}_{\boldsymbol{\theta}}$ is trained well, it will learn to generate data by mimicking the pre-trained model $\mathbf{f}_{\boldsymbol{\theta}_0^-}$ at the boundary. When $t' = 0$, we recover the standard consistency model parameterization Eq. (3). During sampling, as $\mathbf{f}_{\boldsymbol{\theta},\boldsymbol{\theta}_0^-}^{\text{trunc}} = \mathbf{f}_{\boldsymbol{\theta}}$ for all $t \in [t', T]$, we can discard this parameterization and just use $\mathbf{f}_{\boldsymbol{\theta}}$ for generating samples.

To describe the boundary condition, we then partition the support of the time sampling distribution $\psi_t$, i.e., $[t', T]$ into two time ranges: (i) the boundary time region $S_{t'} := \{t \in \mathbb{R} : t' \leq t \leq t' + \Delta_t\}$, and (ii) the consistency training time region $S_{t'}^- \triangleq [t', T] \setminus S_{t'} = \{t \in \mathbb{R} : t' + \Delta_t < t \leq T\}$. To effectively enforce the boundary condition using the first-stage pre-trained model $\mathbf{f}_{\boldsymbol{\theta}_0}$, a nonnegligible amount of $t$'s, sampled from $\psi_t$, must fall within the interval $S_{t'}$. Otherwise the consecutive time steps $t$ and $t - \Delta_t$ in consistency training will mostly be larger or equal to $t'$, limiting the influence of the pre-trained model.

With this time partitioning and our new parameterization, Eq. (4) can be decomposed as follows:

$$\mathcal{L}_{\text{CT}}(\mathbf{f}_{\boldsymbol{\theta},\boldsymbol{\theta}_0^-}^{\text{trunc}}, \mathbf{f}_{\boldsymbol{\theta}^-,\boldsymbol{\theta}_0^-}^{\text{trunc}}) = \underbrace{\int_{t \in S_{t'}} \psi_t(t) \frac{\omega(t)}{\Delta_t} d(\mathbf{f}_{\boldsymbol{\theta}}(\mathbf{x} + t\boldsymbol{\epsilon}, t), \mathbf{f}_{\boldsymbol{\theta}_0^-}(\mathbf{x} + (t - \Delta_t)\boldsymbol{\epsilon}, t - \Delta_t)dt}_{\text{Boundary loss}}$$

$$+ \underbrace{\int_{s \in S_{t'}^-} \psi_t(t) \frac{\omega(t)}{\Delta_t} d(\mathbf{f}_{\boldsymbol{\theta}}(\mathbf{x} + t\boldsymbol{\epsilon}, t), \mathbf{f}_{\boldsymbol{\theta}^-}(\mathbf{x} + (t - \Delta_t)\boldsymbol{\epsilon}, t - \Delta_t)dt}_{\text{Consistency loss}}, \quad (6)$$

where we apply our parameterization in Eq. (5) in the above two time partitions separately, and we drop the expectation over $\mathbf{x} \sim p_{\text{data}}, \boldsymbol{\epsilon} \sim \mathcal{N}(0, \mathbf{I})$ for notation simplicity. Unlike standard consistency training, TCM have two terms: the boundary loss and consistency loss. The boundary loss allows the model to learn from the pre-trained model, preventing collapse to a constant.

Training on the objective (6) can still collapse to a constant if we do not utilize the boundary condition sufficiently by not sampling enough time $t$'s in $S_{t'}$. In particular, this can happen for $\Delta_t$ close to zero when consistency training is near convergence (Song & Dhariwal, 2023; Geng et al., 2024). To prevent this, we design $\psi_t$ to satisfy $\int_{t \in S_{t'}} \psi_t(t)dt > 0$. In other words, we have a strictly positive probability of sampling a point in $S_{t'}$, even when $\Delta_t$ is close to zero. Specifically, we define $\psi_t$ as a mixture of the Dirac delta function $\delta(\cdot)$ at point $t'$ and another distribution $\bar{\psi}_t$:

$$\psi_t(t) = \lambda_b \delta(t - t') + (1 - \lambda_b)\bar{\psi}_t(t), \quad (7)$$

where the weighting coefficient $\lambda_b \in (0, 1)$. $\bar{\psi}_t$ has the support $(t', T]$ and can be instantiated in different ways (e.g., log-normal or log-Student-$t$ distributions); the effect of different $\bar{\psi}_t$ choices is explored in Section 4.4.

By definition, we can see that $\int_{t \in S_{t'}} \psi_t(t)dt \geq \lambda_b$, and $\lambda_b$ controls how significantly we emphasize the boundary condition. Assume that the first-stage consistency model is perfectly trained in $[0, t']$, i.e., $\mathbf{f}_{\boldsymbol{\theta}_0}(\mathbf{x}_t, t) = \mathbf{x}_0$ for all $t \in [0, t']$. If $\mathbf{f}_{\boldsymbol{\theta}}(\mathbf{x}_{t'}, t') \neq \mathbf{f}_{\boldsymbol{\theta}_0}(\mathbf{x}_{t'}, t')$, $\mathbf{f}_{\boldsymbol{\theta}}$ will be penalized by the boundary loss. Minimizing the boundary loss enforces the boundary condition in second-stage model $\mathbf{f}_{\boldsymbol{\theta}}$ (i.e., $\mathbf{f}_{\boldsymbol{\theta}}(\mathbf{x}_{t'}, t') = \mathbf{f}_{\boldsymbol{\theta}_0}(\mathbf{x}_{t'}, t') = \mathbf{x}_0$), while minimizing the consistency loss propagates the boundary condition to the end time (i.e., $\mathbf{f}_{\boldsymbol{\theta}}(\mathbf{x}_T, T) = \mathbf{f}_{\boldsymbol{\theta}}(\mathbf{x}_{t'}, t')$). Consequently, the loss in Eq. (6) effectively guides the model towards the desired solution $\mathbf{f}_{\boldsymbol{\theta}}(\mathbf{x}_T, T) = \mathbf{x}_0$. With the time distribution $\psi_t$ defined in Eq. (7), our training objective becomes

$$\mathcal{L}_{\text{CT}}(\mathbf{f}_{\boldsymbol{\theta},\boldsymbol{\theta}_0^-}^{\text{trunc}}, \mathbf{f}_{\boldsymbol{\theta}^-,\boldsymbol{\theta}_0^-}^{\text{trunc}}) \approx \lambda_b \underbrace{\frac{\omega(t')}{\Delta_{t'}} d(\mathbf{f}_{\boldsymbol{\theta}}(\mathbf{x} + t'\boldsymbol{\epsilon}, t'), \mathbf{f}_{\boldsymbol{\theta}_0^-}(\mathbf{x} + (t' - \Delta_{t'})\boldsymbol{\epsilon}, t' - \Delta_{t'}))}_{\text{Boundary loss}:=\mathcal{L}_B(\mathbf{f}_{\boldsymbol{\theta}}, \mathbf{f}_{\boldsymbol{\theta}_0^-})} \quad (8)$$

$$+ (1 - \lambda_b) \mathbb{E}_{\bar{\psi}_t}[\underbrace{\frac{\omega(t)}{\Delta_t} d(\mathbf{f}_{\boldsymbol{\theta}}(\mathbf{x} + t\boldsymbol{\epsilon}, t), \mathbf{f}_{\boldsymbol{\theta}^-}(\mathbf{x} + (t - \Delta_t)\boldsymbol{\epsilon}, t - \Delta_t))}_{\text{Consistency loss}:=\mathcal{L}_C(\mathbf{f}_{\boldsymbol{\theta}}, \mathbf{f}_{\boldsymbol{\theta}^-})}]. \quad (9)$$

---

**Algorithm 1** Truncated Consistency Training

---

1: **Standard consistency training**
2: $\boldsymbol{\theta}_0 \leftarrow \arg\min_{\hat{\boldsymbol{\theta}}} \mathcal{L}_{\mathrm{CT}}(\mathbf{f}_{\hat{\boldsymbol{\theta}}}, \mathbf{f}_{\hat{\boldsymbol{\theta}}^-})$          ▷ Optimize consistency training loss for the regular model
3: **Truncated training**
4: $N_B \leftarrow \lfloor B\rho \rfloor$          ▷ Number of boundary samples
5: **for** each training iteration **do**
6:      $\mathbf{x}_1, ..., \mathbf{x}_B \sim p_{\mathrm{data}}, \boldsymbol{\epsilon}_1, ..., \boldsymbol{\epsilon}_B \sim \mathcal{N}(\mathbf{0}, \mathbf{I})$
7:      Set $t_1, ..., t_{N_B}$ to $t'$, and $t_{N_B+1}, ..., t_B \sim \bar{\psi}_t$
8:      Compute $\sum_{i=1}^{N_B} (\mathcal{L}_B)_i (\mathbf{f}_{\boldsymbol{\theta}}, \mathbf{f}_{\boldsymbol{\theta}_0^-})$ using Eq. (8) with $(\mathbf{x}_i, \boldsymbol{\epsilon}_i, t_i)$ for $i = 1, ..., N_B$
9:      Compute $\sum_{j=N_B+1}^{B} (\mathcal{L}_C)_j (\mathbf{f}_{\boldsymbol{\theta}}, \mathbf{f}_{\boldsymbol{\theta}^-})$ using Eq. (9) with $(\mathbf{x}_j, \boldsymbol{\epsilon}_j, t_j)$ for $j = N_B + 1, ..., B$
10:      Compute $\nabla_{\boldsymbol{\theta}} \mathcal{L}_{\mathrm{TCM}}$ using Eq. (11)
11:      Update $\boldsymbol{\theta}$ using the computed gradient
12: **end for**

---

where the approximation in Eq. (8) holds when $\Delta_t$ is sufficiently small (which is true for the truncated training stage). Please see Appendix E for the detailed derivation. For simplicity of notation, we relax the above objective by absorbing the $(1 - \lambda_b)$ factor into $\lambda_b$ and express our final training loss as:

$$\mathcal{L}_{\mathrm{TCM}} := w_b \mathcal{L}_B(\mathbf{f}_{\boldsymbol{\theta}}, \mathbf{f}_{\boldsymbol{\theta}_0^-}) + \mathcal{L}_C(\mathbf{f}_{\boldsymbol{\theta}}, \mathbf{f}_{\boldsymbol{\theta}^-}), \tag{10}$$

where $w_b = \lambda_b / (1 - \lambda_b)$ is a tunable hyperparameter that controls the weighting of the boundary loss. To estimate the two losses, we partition each mini-batch of size $B$ into two subsets. The boundary loss $L_B$ is estimated using $N_B = \lfloor B\rho \rfloor$ samples, where $\rho \in (0, 1)$ is a hyperparameter controlling the allocation of samples. The consistency loss $L_C$ is estimated with the remaining $B - N_B$ samples. Increasing $\rho$ reduces the variance of the boundary loss gradient estimator but increases the variance of the consistency loss gradient estimator, and vice versa. The final mini-batch loss is as follows:

$$\mathcal{L}_{\mathrm{TCM}} \approx \frac{w_b}{N_B} \sum_{i=1}^{N_B} \nabla_{\boldsymbol{\theta}} (\mathcal{L}_B)_i (\mathbf{f}_{\boldsymbol{\theta}}, \mathbf{f}_{\boldsymbol{\theta}_0^-}) + \frac{1}{B - N_B} \sum_{j=N_B+1}^{B} \nabla_{\boldsymbol{\theta}} (\mathcal{L}_C)_j (\mathbf{f}_{\boldsymbol{\theta}}, \mathbf{f}_{\boldsymbol{\theta}^-}), \tag{11}$$

where $(\mathcal{L}_B)_i$ and $(\mathcal{L}_C)_j$ are the boundary loss and the consistency loss at the $i$-th sample from $\delta(t - t')$ and the $j$-th sample from $\bar{\psi}_t$, respectively. We provide the training algorithm in Algorithm 1.

## 4 EXPERIMENTS

In this section, we evaluate TCM on standard image generation benchmarks and compare it against state-of-the-art generative models. We begin by detailing the experimental setup in Sec. 4.1. We then study the behavior of denoising FID and its impact on generation FID in Sec. 4.2. We benchmark TCM against a variety of existing methods in Sec. 4.3, and provide detailed analysis on various design choices in Sec. 4.4.

### 4.1 SETUP

We evaluate TCM on the CIFAR-10 (Krizhevsky et al., 2009) and ImageNet $64\times64$ (Deng et al., 2009) datasets. We consider the unconditional generation task on CIFAR-10 and class-conditional generation on ImageNet $64\times64$. We measure sample quality with Fréchet Inception Distance (FID) (Heusel et al., 2017) (lower is better), as is standard in the literature.

For consistency training in TCM, we mostly follow the hyperparameters in ECT (Geng et al., 2024), including the discretization curriculum and continuous-time training schedule. For all experiments, we choose a dividing time $t' = 1$ and set $\bar{\psi}_t$ to the log-Student-$t$ distribution. We use $w_b = 0.1$ and $\rho = 0.25$ for the boundary loss. We discuss these choices in Sec. 4.4. In line with Geng et al. (2024), we initialize the model with the pre-trained EDM (Karras et al., 2022) / EDM2 (Karras et al., 2024) for CIFAR-10 / ImageNet $64 \times 64$, respectively. On CIFAR-10, we use a batch size of 512 and 1024 for the first and the second stage, respectively. On ImageNet with EDM2-S architecture, we use a batch size of 2048 and 1024 for the first and the second stage, respectively. For EDM2-XL, to save compute, we initialize the truncated training stage with the pre-trained checkpoint from the ECM work (Geng et al., 2024) that performs the standard consistency training, and conduct the second-stage training with a batch size of 1024. Please see Appendix F for more training details.

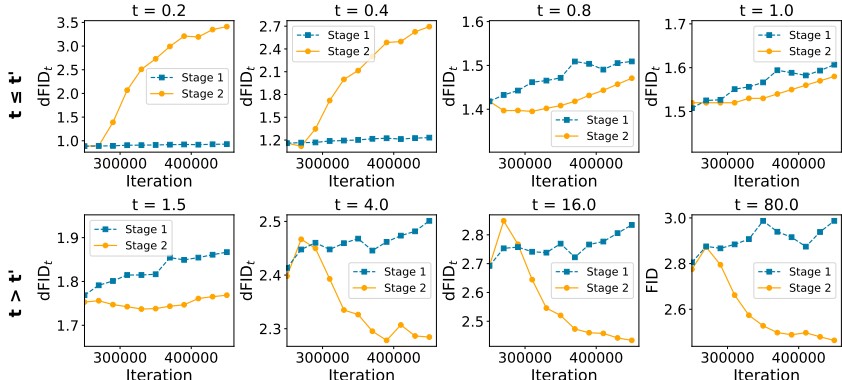

Figure 3: Denoising FID (dFID) for continuation of standard consistency training at later iterations (Stage 1) and TCM model (Stage 2) at various $t$s on CIFAR-10 during the course of training. For TCM, we set the dividing time $t' = 1$. We can see, in the second stage, the dFID exhibits a dramatic increase at times below the dividing time $t'$, while the dFID at times above $t'$ and FID at $t = T$ continue to improve. Notably, the rate of dFID in the truncated stage increase at earlier times is significantly faster compared to standard consistency training, suggesting a more efficient "forgetting" of the denoising tasks.

## 4.2 TRUNCATED TRAINING ALLOCATES CAPACITY TOWARD GENERATION

Our proposed TCM aims to explictly reallocate network capacity towards generation by de-emphasizing denoising tasks at smaller $t$'s. Empirical analysis in Fig. 3 further characterizes this behavior, showing a rapid increase in dFIDs at smaller $t$'s below the threshold $t'$ during the truncated training stage. Conversely, dFIDs continue to decrease at larger $t$'s. In addition, TCM exhibit a more pronounced "forgetting" of the denoising task compared to consistency training (Fig. 2) at earlier times. For instance, dFID at $t = 0.2$ increases up to 3.5 in the truncated training, whereas it remains below 1 in the standard consistency training. TCM also significantly accelerate the process of forgetting the denoising tasks at these earlier times, achieving a substantially improved generation FID. This suggests that by explicitly controlling the training time range, the neural network can effectively shift its capacity towards generation.

Fig. 1(b) demonstrates how this reallocation of network capacity directly translates to improved sample quality and training stability. For CIFAR-10 / ImageNet $64 \times 64$, the truncated training stage (Stage 2) is initialized from the Stage 1 model at 250K / 150K iterations, respectively. We can see that the truncated training improves FID over the consistency training on the two datasets. Moreover, we find that the

Table 1: FID, NFE and # param. on CIFAR-10. Bold indicates the best result for each category and NFE.

| Method | NFE | FID | # param. (M) |
|---|---|---|---|
| **Diffusion models** | | | |
| EDM (Karras et al., 2022) | 35 | 1.97 | 55.7 |
| PFGM++ (Xu et al., 2023b) | 35 | **1.91** | 55.7 |
| DDPM (Ho et al., 2020) | 1000 | 3.17 | 35.7 |
| LSGM (Vahdat et al., 2021) | 147 | 2.10 | 475 |
| **Consistency models** | | | |
| **1-step** | | | |
| iCT (Song & Dhariwal, 2023) | 1 | 2.83 | 56.4 |
| iCT-deep (Song & Dhariwal, 2023) | 1 | 2.51 | 112 |
| CTM (Kim et al., 2023) (*w/o GAN*) | 1 | 5.19 | 55.7 |
| ECM (Geng et al., 2024) | 1 | 3.60 | 55.7 |
| TCM (*ours*) | 1 | **2.46** | 55.7 |
| **2-step** | | | |
| iCT (Song & Dhariwal, 2023) | 2 | 2.46 | 56.4 |
| iCT-deep (Song & Dhariwal, 2023) | 2 | 2.24 | 112 |
| ECM (Geng et al., 2024) | 2 | 2.11 | 55.7 |
| TCM (*ours*) | 2 | **2.05** | 55.7 |
| **Variational score distillation** | | | |
| DMD (Yin et al., 2024b) | 1 | 3.77 | 55.7 |
| Diff-Instruct (Luo et al., 2024) | 1 | 4.53 | 55.7 |
| SiD (Zhou et al., 2024) | 1 | **1.92** | 55.7 |
| **Knowledge distillation** | | | |
| KD (Luhman & Luhman, 2021) | 1 | 9.36 | 35.7 |
| DSNO (Zheng et al., 2022a) | 1 | **3.78** | 65.8 |
| TRACT (Berthelot et al., 2023) | 1 | **3.78** | 55.7 |
| | 2 | **3.32** | 55.7 |
| PD (Salimans & Ho, 2022) | 1 | 9.12 | 60.0 |
| | 2 | 4.51 | 60.0 |

truncated training is more stable than the original consistency training, as their ImageNet FID blows up after 150K iterations, while TCM continues to improve FID from 2.83 to 2.46, showcasing its robustness (See Figure 7 for more analysis).

## 4.3 TCM IMPROVES THE SAMPLE QUALITY OF CONSISTENCY MODELS

To demonstrate the effectiveness of TCM, we compare our method with three lines of works that distill diffusion models to one or two steps: (i) *consistency models* (Song & Dhariwal, 2023; Kim et al., 2023; Geng et al., 2024) that distills the PF ODE mapping in a simulation-free manner; (ii) *variational score distillation* (Yin et al., 2024b; Luo et al., 2024; Zhou et al., 2024) that performs distributional matching by utilizing the score of pre-trained diffusion models; (iii) *knowledge distillation* (Luhman & Luhman, 2021; Zheng et al., 2022a; Berthelot et al., 2023; Salimans & Ho, 2022) that distill the PF ODE through off-line or on-line simulation using the pre-trained diffusion models. We exclude the methods that additionally use the GAN loss, which causes more training difficulties, for fair comparison.

**Results.** In Table 1 and Table 2, we report the sample quality measured by FID and the sampling speed measured by the number of function evaluations (NFE), on CIFAR-10 and ImageNet-64×64, respectively. We mostly borrow the baseline results from the original papers. We also include the number of model parameters. Our main findings are: **(1) TCM significantly outperforms improved Consistency Training (iCT) (Song & Dhariwal, 2023), the state-of-the-art consistency model, across datasets, number of steps and network sizes**. For example, TCM improves the one-step FID from 2.83 / 4.02

Table 2: FID, NFE and # param. on ImageNet $64 \times 64$. Dotted lines separate results by # param. Bold indicates the best result for each category and NFE.

| Method | NFE | FID | # param. (M) |
|---|---|---|---|
| **Diffusion models** | | | |
| EDM2-S (Karras et al., 2024) | 63 | 1.58 | 280 |
| EDM2-XL (Karras et al., 2024) | 63 | 1.33 | 1119 |
| **Consistency models** | | | |
| **1-step** | | | |
| iCT (Song & Dhariwal, 2023) | 1 | 4.02 | 296 |
| iCT-deep (Song & Dhariwal, 2023) | 1 | 3.25 | 592 |
| ECM (Geng et al., 2024) (*EDM2-S*) | 1 | 4.05 | 280 |
| TCM (*ours*; *EDM2-S*) | 1 | **2.88** | 280 |
| MultiStep-CD (Heek et al., 2024) | 1 | 3.20 | 1200 |
| ECM (Geng et al., 2024) (*EDM2-XL*) | 1 | 2.49 | 1119 |
| TCM (*ours*; *EDM2-XL*) | 1 | **2.20** | 1119 |
| **2-step** | | | |
| iCT (Song & Dhariwal, 2023) | 2 | 3.20 | 296 |
| iCT-deep (Song & Dhariwal, 2023) | 2 | 2.77 | 592 |
| ECM (Geng et al., 2024) (*EDM2-S*) | 2 | 2.79 | 280 |
| TCM (*ours*; *EDM2-S*) | 2 | **2.31** | 280 |
| MultiStep-CD (Heek et al., 2024) | 2 | 1.90 | 1200 |
| ECM (Geng et al., 2024) (*EDM2-XL*) | 2 | 1.67 | 1119 |
| TCM (*ours* ; *EDM2-XL*) | 2 | **1.62** | 1119 |
| **Variational score distillation** | | | |
| DMD2 w/o GAN (Yin et al., 2024a) | 1 | 2.60 | 296 |
| Diff-Instruct (Luo et al., 2024) | 1 | 5.57 | 296 |
| EMD-16 (Xie et al., 2024) | 1 | 2.20 | 296 |
| Moment Matching (Salimans et al., 2024) | 1 | 3.00 | 400 |
| | 2 | 3.86 | 400 |
| SiD (Zhou et al., 2024) | 1 | **1.52** | 296 |
| **Knowledge distillation** | | | |
| DSNO (Zheng et al., 2022a) | 1 | 7.83 | 329 |
| TRACT (Berthelot et al., 2023) | 1 | 7.43 | 296 |
| | 2 | **4.97** | 296 |
| PD (Salimans & Ho, 2022) | 1 | 15.4 | 296 |
| | 2 | 8.95 | 296 |

in iCT to 2.46 / 2.88, on CIFAR-10 / ImageNet. Further, TCM's one-step FID even rivals iCT's two-step FID on both datasets. When using EDM2-S model, TCM also surpasses iCT-deep, which uses $2\times$ deeper networks, in both one-step (2.88 vs 3.25) and two-step FIDs (2.31 vs 2.77) on ImageNet. **(2) TCM beats all the knowledge distillation methods and performs competitively to variational score distillation methods.** Note that TCM do not need to train additional neural networks as in VSD methods, or to run simulation as in knowledge distillation methods. **(3) Two-step TCM performs comparably to the multi-step EDM (Karras et al., 2022), the state-of-the-art diffusion model**. For example, when both using the same EDM network, two-step TCM obtains a FID of 2.05 on CIFAR-10, which is close to 1.97 in EDM with 35 sampling steps. We further provide the uncurated one-step and two-step generated samples in Fig. 5. Please see Appendix G for more samples.

## 4.4 ANALYSES OF DESIGN CHOICES

**Time sampling distribution $\bar{\psi}_t$.** We explore various time sampling distributions $\bar{\psi}_t$ supported on $[t', T]$, and find that the truncated log-Student-$t$ distribution works best (*i.e.*, $\ln(t)$ follows Student-$t$ distribution). The Student-$t$ distribution, being heavier-tailed than the Gaussian distribution employed in previous consistency training (Song & Dhariwal, 2023; Geng et al., 2024), inherently allocates more probability mass towards larger $t$'s. This aligns with the motivation of TCM, which emphasizes enhancing generation capabilities at later times. The degree of freedom $\nu$ effectively controls the thickness of the tail, with the Student-$t$ distribution converging to a Gaussian distribution as $\nu \to \infty$. Figure 4a shows the shape of $\bar{\psi}_t$ with varying standard deviation $\sigma$ and the degree of freedom $\nu$ in three cases: (1) heavy-tailed and a low probability mass around small $t$'s ($\sigma = 2, \nu = 10000$),

Table 3: CIFAR-10 FID when varying the dividing time $t'$.

| $t'$ value | 0.17 | 0.8 | 1.0 | 1.5 |
|---|---|---|---|---|
| FID | 2.70 | 2.69 | **2.56** | 2.79 |

Table 4: CIFAR-10 FID for different training stages.

| | Stage 1 | Stage 2 | Stage 3 |
|---|---|---|---|
| FID | 2.77 | 2.46 | 2.46 |

(2) heavy-tailed and a high probability mass around small $t$'s ($\sigma = 0.2, \nu = 0.01$), (2) light-tailed ($\sigma = 0.2, \nu = 2$). From Fig. 4b, we observe that the log-Student-$t$ distribution with $\sigma = 0.2, \nu = 0.01$ is the best among the three. Hence we use $\sigma = 0.2, \nu = 0.01$ in all the experiments.

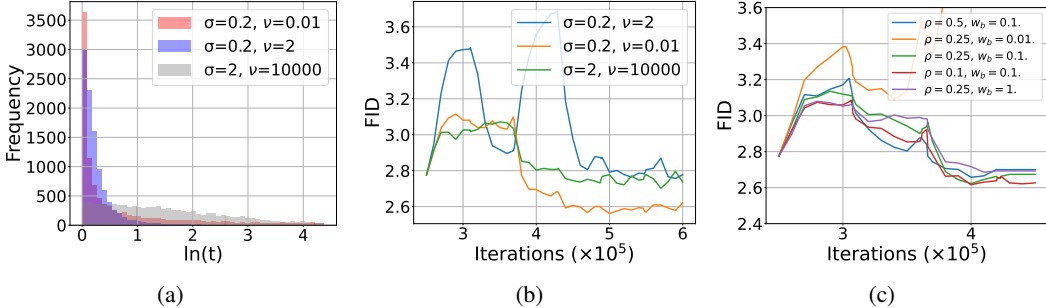

(a)                                 (b)                                 (c)

Figure 4: **(a)** Comparison of Student-$t$ distributions with different standard deviations $\sigma$ and degree of freedom $\nu$. **(b)** FID evolution on CIFAR-10 for different $\sigma$ and $\nu$. $w_b = 0.1, \rho = 0.25, t' = 1$, and a batch size of 128 are used for all plots. **(c)** Effect of $\rho$ and $w_b$ on the FID on CIFAR-10. We use a batch size of 128. $t'$ is set to 1.

**Strength for boundary loss.** Figure 4c shows the effect of two key hyper-parameters that control the strength of imposing boundary condition in the TCM objective (Eq. 10). We observe that the FID is relatively stable with a wide range of $\rho$ and $w_b$ (from 0.1 to 0.5 for $\rho$ and from 0.1 to 1 for $w_b$). However, when using a very small weight for the boundary loss ($w_b = 0.01$), FID explodes as the model fails to maintain the boundary condition. Thus, we use $\rho = 0.25, w_b = 0.1$ in all the experiments.

**Dividing time $t'$.** The boundary $t'$ ideally represents the point where the task in the PF ODE transitions from denoising to generation. However, this transition is gradual, and there is no single definitive point. Fig. 2b suggests this transition occurs roughly between $t' = 0.8$ and $t' = 1.5$, where we observe a change in dFID behavior: it primarily deteriorates during training before this range, but then stabilizes afterwards (more indicative of a generation task). Based on this analysis, we experimented with multiple $t'$ values around this range. Table 3 shows that $t' = 1$ provides the best results among the choices. Note that here we use a batch size of 128, while it is 1024 in our default setting.

**Are two stages enough?** A natural question is whether we can extend our two-stage training procedures to three or more stages by gradually increasing $t'$. However, recall that our methodology was motivated by the fact that in the first-stage training (standard consistency training), we observe increasing dFIDs at smaller $t$ values of the training range, as seen in Fig. 2. This trade-off is notably absent in the second stage over the time range $[t', T]$, as seen in Fig. 3. This suggests that during the second stage, the model tackles tasks that are more or less similar to generation, and introducing another truncated training stage may not yield further gains. In support of this hypothesis, we implement the third stage where the dividing time is $t' = 4$, but do not observe improvement, as shown in Table 4. We also consider adding an intermediate training stage between stage 1 and stage 2 that finetunes $\mathbf{f}_{\theta_0}$ in the time range $(0, t')$ but it produces a slightly worse performance, which we discuss in Appendix C.

## 5    RELATED WORK

**Consistency models.** Song et al. (2023) first proposed consistency models as a new class of generative models that synthesize samples with a single network evaluation. Later, Song & Dhariwal (2023); Geng et al. (2024) presented a set of improved techniques to train consistency models for better sample quality. Luo et al. (2023) introduced latent consistency models (LCM) to accelerate the sampling of latent diffusion models. Kim et al. (2023) proposed consistency trajectory models (CTM) that generalize consistency models by enabling the prediction between any two intermediate points on the same PF ODE trajectory. The training objective in CTM becomes more challenging

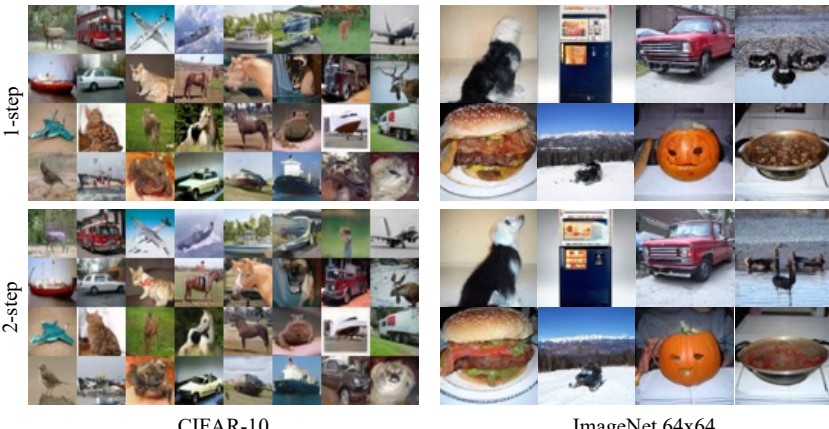

CIFAR-10                    ImageNet 64x64

Figure 5: Uncurated one-step (top) and two-step (bottom) generated samples from TCM (EDM) on CIFAR-10 and TCM (EDM2-XL) on ImageNet 64×64, respectively.

than standard consistency models that only care about the mapping from intermediate points to the data endpoints. Heek et al. (2024) proposed multistep consistency models that divide the PF ODE trajectory into multiple segments to simplify the consistency training objective. They train the consistency models in each segment separately, and need multiple steps to generate a sample. Similar direction is also explored in Wang et al. (2024). Ren et al. (2024) combined CTM with progressive distillation (Salimans & Ho, 2022), by performing segment-wise consistency distillation where the number of ODE trajectory segments progressively reduces to one. Similar to CTM, it relies on the adversarial loss (Goodfellow et al., 2014) to achieve good performance.

**Fast sampling of diffusion models.** While a line of work aims to accelerate diffusion models via fast numerical solvers for the PF-ODE (Lu et al., 2022; Karras et al., 2022; Liu et al., 2022a; Xu et al., 2023a), they usually still require more than $10$ steps. To achieve low-step or even one-step generation, besides consistency models, other training-based methods have been proposed from three main perspectives: (i) *Knowledge distillation*, which first used the pre-trained diffusion model to generate a dataset of noise and image pairs, and then applied it to train a single-step generator (Luhman & Luhman, 2021; Zheng et al., 2022a). Progressive distillation (Salimans & Ho, 2022; Meng et al., 2023) iteratively halves the number of sampling steps required, without needing an offline dataset. (ii) *Variational score distillation*, which aims to match the distribution of the student and teacher output via an approximate (reverse) KL divergence (Yin et al., 2024b;a; Xie et al., 2024), implicit score matching (Zhou et al., 2024) or moment matching (Salimans et al., 2024). (iii) *Adversarial distillation*, which leverages the adversarial training to fine-tune pre-trained diffusion models into a few-step generator (Sauer et al., 2023; 2024; Lin et al., 2024; Xu et al., 2024). Compared with these training-based diffusion acceleration methods, our method is most memory and computation efficient.

**Truncated training of diffusion models.** Balaji et al. (2022) propose to train different diffusion models for each time step range. Since consistency models solve a more difficult task (learning to integrate PF-ODE) than diffusion models (learning the drift of PF-ODE), they can benefit more from such a strategy but also require a specific parameterization (Eq. (5)) to satisfy the boundary condition. Zheng et al. (2022b) use GANs to generate the noised data and use diffusion models to map them to clean data. Different from ours, they train diffusion models on the first half of the interval.

## 6   CONCLUSION

We have introduced a truncated consistency training method that significantly enhances the sample quality of consistency models. To generalize consistency models to the truncated time range, we have proposed a new parameterization of the consistency function and a two-stage training process that explicitly allocates network capacity towards generation. We also discussed about our design choices arising from the new training paradigm. Our approach achieves superior performance compared to state-of-the-art consistency models, as evidenced by improved one-step and two-step FID scores across different datasets and network sizes. Notably, these improvements are achieved while utilizing similar or even smaller network architectures than baselines.

## 7 REPRODUCIBILITY STATEMENT

We provide sufficient details for reproducing our method in the main paper and also in the Appendix. F, including a pseudo code of the training algorithm, model initialization and architecture, model parameterization, learning rate schedules, time step sampling procedures, and other training details. We also specify hyperparameter choices like the dividing time $t'$, boundary loss weight $w_b$, and boundary ratio $\rho$. Additionally, we discuss the computational costs of our method compared to standard consistency training. For evaluation, we describe our sampling procedure for both one-step and two-step generation.

## 8 ETHICS STATEMENT

This paper raises similar ethical concerns to other papers on deep generative models. Namely, such models can be (and have been) used to generate harmful content, such as disinformation and violent imagery. We advocate for the responsible deployment of such models in practice, including guardrails to reduce the risk of producing harmful content. The design of these protections is orthogonal to our work. Other ethical concerns may arise regarding the significant resource costs required to train and use deep generative models, including energy and water usage. This work increases the training cost of consistency models, but it also enables the models to be run with only 1 NFE and requires smaller neural network architectures, both of may which reduce inference-time costs relative to other diffusion-based models. Nonetheless, the environmental impact of training and deploying deep generative models remains an important limitation.

## ACKNOWLEDGMENTS

We thank Zhengyang Geng for helpful feedback on reproducing ECM. This work was supported in part by the National Science Foundation through RINGS grant 2148359. GF also acknowledges the generous support of the Sloan Foundation, Intel, Bosch, and Cisco.

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

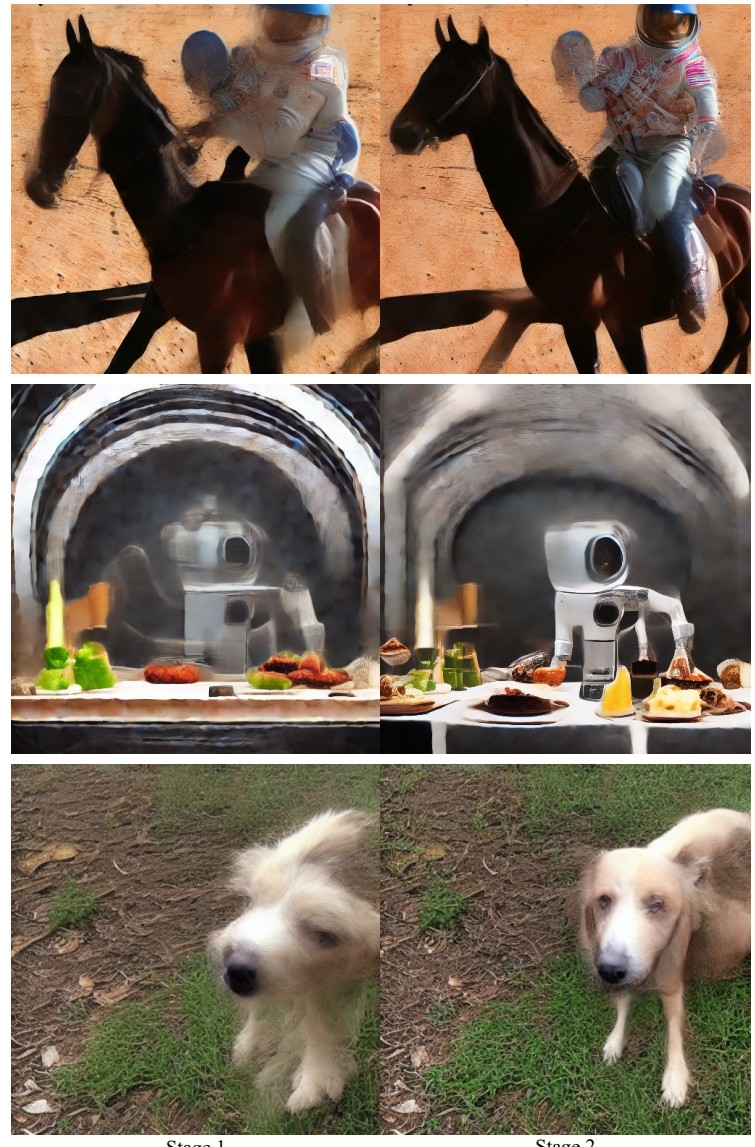

Stage 1         Stage 2

Figure 6: One-step text-to-image generation results of the standard consistency model (stage 1) and TCM (stage 2).

## A  LIMITATION

TCM introduces an additional training stage on top of the standard consistency model training. Compared to the standard consistency training, the truncated training requires a slight increase in per-iteration training time due to the additional boundary loss in Eq. (10). Standard consistency training necessitates two forward passes per training iteration, while our parameterization (Eq. 5) requires three. Also, the truncated training incurs a minor additional memory cost as we need to maintain a pre-trained consistency model (in evaluation mode) for the boundary loss. We observe that on ImageNet $64\times64$ with EDM2-S, TCMs have an 18% increase in training time per iteration and an 15% increase in memory cost. Moreover, the one-step sample quality of TCM still has a considerable performance gap from the diffusion models with a large NFE, but we believe this is an important step toward closing the gap.

## B TEXT-TO-IMAGE RESULTS

Table 5: Zero-shot FID scores on MSCOCO dataset measured with 30k generated samples.

|  | Stage 1 | Stage 2 |
|---|---|---|
| FID $\downarrow$ | 18.32 | **15.58** |

To show the scalability of our method, we train TCM on COYO dataset [1], using consistency distillation with a fixed classifier-free guidance (Ho & Salimans, 2022) scale of 6. We initialize our models with stable diffusion (Rombach et al., 2022) 1.5. We use a batch size of 512 for a quick validation, though using a larger batch size ($\geq 1,024$) is standard (Liu et al., 2023; Yin et al., 2024a) and would lead to better generative performance. For the first stage, we train for 80,000 iterations (after which FID starts to increase), and in the second stage, we additionally train for another 200,000 iterations. We provide visual comparison between the standard consistency model and TCM in Fig. 6. Captions used are: "A photo of an astronaut riding a horse on Mars", "Robot serving dinner, metallic textures, futuristic atmosphere, high-tech kitchen, elegant plating, intricate details, high quality, misc-architectural style, warm and inviting lighting", and "A photo of a dog" for each row.

We also measure the FID on MSCOCO dataset (Lin et al., 2014) in Table. 5. We see that TCM achieves a better FID than the standard consistency model (the first stage).

## C ADDITIONAL EXPERIMENTS

Fig. 7 shows that the gradient spikes during the first stage training while the second stage training is relatively smooth. We hypothesize that the truncated training is more stable because it is less affected by the biased gradient norms across different $t$.

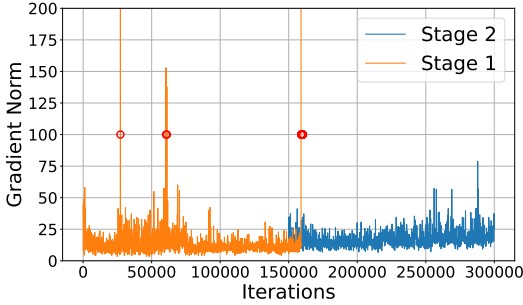

Figure 7: Gradient norm evolution during the first and second stage training on ImageNet $64 \times 64$ (corresponding to Fig. 1(b)). The red circles indicate where the gradient norms are larger than 100. Stage 1 training blows up after the last few gradient spikes. It shows that the truncated consistency training is more stable than the standard consistency training.

Fig. 8 shows the dFID$_t$ evolution during the standard consistency training. We see that dFIDs at larger $t$'s start from larger values and converges more slowly.

**Adding an intermediate training stage** In our parameterization Eq. (5), we only use the pre-trained model $\mathbf{f}_{\boldsymbol{\theta}_0}$ in $[0, t')$. One may wonder if we can fine-tune $\mathbf{f}_{\boldsymbol{\theta}_0}$ on the truncated time range $[0, t')$ to provide a better boundary condition for the truncated training. We find that although doing so improved the dFID$_{t'}$ of $\mathbf{f}_{\boldsymbol{\theta}_0}$ from 1.51 to 1.43, it led to a worse final FID of $>2.7$ for the truncated consistency model, regardless of whether we initialized $\mathbf{f}_{\boldsymbol{\theta}}$ with the pre-trained model or the fine-tuned model. In contrast, our proposed method achieved an FID of 2.61 with the same hyperparameters. We hypothesize that fine-tuning the pre-trained model on the truncated time range $[0, t')$ makes the model $\mathbf{f}_{\boldsymbol{\theta}_0}$ forget about how the early mappings properly propagate to the later mappings in the range of $[t', T]$. This may hinder the learnability of its mapping at the boundary time, making it harder for $\mathbf{f}_{\boldsymbol{\theta}}$ to transfer the knowledge learned in $\mathbf{f}_{\boldsymbol{\theta}_0}$ to its generation capability.

---

[1] https://github.com/kakaobrain/coyo-dataset

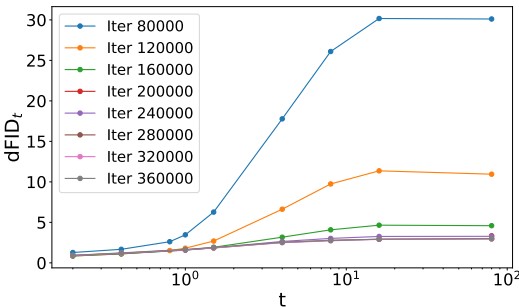

Figure 8: Evolution of the denoising FIDs ($\text{dFID}_t$) at different times $t$'s during standard consistency training for different iterations. For $t \in (1, 10)$, $\text{dFID}_t$ has different convergence speeds while in both small times ($t < 1$) and large 's ($t > 10$), $\text{dFID}_t$ converges with a more similar speed.

## D BACKGROUND ON CONSISTENCY MODELS

Most of this part has been introduced by previous works (Song et al., 2023; Song & Dhariwal, 2023). Here, we introduce the background of consistency models, in particular the relationship between consistency training and consistency distillation, for completeness.

### D.1 DEFINITION OF CONSISTENCY FUNCTION

#### D.1.1 PROBABILITY FLOW ODE

The probability flow ODE (PF ODE) of Karras et al. (2022) is as follows:

$$d\mathbf{x}_t = -t\mathbf{s}_t(\mathbf{x}_t)dt, \tag{12}$$

where $\mathbf{s}_t(\mathbf{x}_t)$ is the score function at time $t \in [0, T]$. To draw samples from the data distribution $p_{\text{data}}$, we initialize $\mathbf{x}_T$ with a sample from $\mathcal{N}(0, T^2\mathbf{I})$ and solve the PF ODE backward in time. The solution $\mathbf{x}_0 = \mathbf{x}_T + \int_T^0 (-t\mathbf{s}_t(\mathbf{x}_t))dt$ is distributed according to $p_{\text{data}}$.

#### D.1.2 CONSISTENCY FUNCTION

Integrating the PF ODE using numerical solvers is computationally expensive. Consistency function instead directly outputs the solution of the PF ODE starting from any $t \in [0, T]$. The consistency function $\mathbf{f}$ satisfies the following two properties:

1. $\mathbf{f}(\mathbf{x}_0, 0) = \mathbf{x}_0$.
2. $\mathbf{f}(\mathbf{x}_t, t) = \mathbf{f}(\mathbf{x}_s, s) \; \forall(s, t) \in [0, T]^2$.

The first condition can be trivially satisfied by setting $\mathbf{f}(\mathbf{x}, t) = c_{\text{out}}(t)\mathbf{F}(\mathbf{x}, t) + c_{\text{skip}}(t)\mathbf{x}$ where $c_{\text{out}}(0) = 0$ and $c_{\text{skip}}(0) = 1$ following EDM (Karras et al., 2022). The second condition can be satisfied by optimizing the following objective:

$$\min_{\mathbf{f}} \mathbb{E}_{s,t,\mathbf{x}_t}[d(\mathbf{f}(\mathbf{x}_t, t), \mathbf{f}(\mathbf{x}_s, s))], \tag{13}$$

where $d$ is a function satisfying:

1. $d(\mathbf{x}, \mathbf{y}) = 0 \iff \mathbf{x} = \mathbf{y}$.
2. $d(\mathbf{x}, \mathbf{y}) \geq 0$.
3. $\frac{\partial d(\mathbf{x}, \mathbf{y})}{\partial \mathbf{y}}|_{\mathbf{y}=\mathbf{x}} = 0$
4. $\frac{\partial \mathbf{f}_{\boldsymbol{\theta}}}{\partial \boldsymbol{\theta}}$ and $\frac{\partial d}{\partial \mathbf{y}^2}$ are well-defined and bounded.

## D.2 Consistency distillation

### D.2.1 Objective

In practice, Song et al. (2023) consider the following objective instead:

$$\min_{\boldsymbol{\theta}} \; \mathbb{E}_{t,\mathbf{x}_t}[d(\mathbf{f}_{\boldsymbol{\theta}}(\mathbf{x}_t, t), \mathbf{f}_{\boldsymbol{\theta}^-}(\mathbf{x}_{t-\Delta_t}, t - \Delta_t))], \tag{14}$$

where we parameterize the consistency function $\mathbf{f}$ with a neural network $\mathbf{f}_{\boldsymbol{\theta}}$, and $0 < \Delta_t < t$. Here, $\mathbf{f}_{\boldsymbol{\theta}^-}$ is the identical network with stop gradients applied and is called *teacher*. Since $\Delta_t > 0$, the teacher always receives the less noisy input, and the student $\mathbf{f}_{\boldsymbol{\theta}}$ is trained to mimic the teacher. Optimizing Eq. (14) requires computing $\mathbf{x}_{t-\Delta_t}$, which we can be approximated using one step of Euler's solver:

$$\mathbf{x}_{t-\Delta_t} = \mathbf{x}_t + \int_t^{t-\Delta_t} (-u\mathbf{s}_u(\mathbf{x}_u))du \approx \mathbf{x}_t + t\mathbf{s}_t(\mathbf{x}_t)\Delta_t. \tag{15}$$

When $\mathbf{s}_t(\mathbf{x}_t)$ is approximated by a pre-trained score network, Eq. (14) becomes the consistency distillation objective in Song et al. (2023). If $\Delta_t$ is sufficieintly small, the approximation in Eq. (15) is quite accurate, making $\mathcal{L}_{\text{CD}}$ a good approximation of Eq. (14). The precision of the approximation depends on $\Delta_t$ and also the trajectory curvature of the PF ODE.

### D.2.2 Gradient when $\Delta_t \to 0$

Let us rewrite Eq. (14) as follows:

$$\mathbb{E}_{t,\mathbf{x}_t}[d(\mathbf{f}_{\boldsymbol{\theta}}(\mathbf{x}_t, t), \mathbf{f}_{\boldsymbol{\theta}^-}(\mathbf{x}_{t-\Delta_t}, t - \Delta_t))] \tag{16}$$

$$= \mathbb{E}_{t,\mathbf{x}_t}[d(\mathbf{f}_{\boldsymbol{\theta}}(\mathbf{x}_t, t), \underbrace{\mathbf{f}_{\boldsymbol{\theta}}(\mathbf{x}_t, t)}_{\mathbf{y}} + \underbrace{\mathbf{f}_{\boldsymbol{\theta}^-}(\mathbf{x}_{t-\Delta_t}, t - \Delta_t) - \mathbf{f}_{\boldsymbol{\theta}}(\mathbf{x}_t, t)}_{\Delta\mathbf{y}})] \tag{17}$$

$$= \mathbb{E}_{t,\mathbf{x}_t}[d(\mathbf{f}_{\boldsymbol{\theta}}(\mathbf{x}_t, t), \mathbf{f}_{\boldsymbol{\theta}}(\mathbf{x}_t, t)) + \frac{\partial d}{\partial \mathbf{y}}\Delta\mathbf{y} + \frac{1}{2}(\Delta\mathbf{y})^T \frac{\partial^2 d}{\partial \mathbf{y}^2}\Delta\mathbf{y} + O(||\Delta\mathbf{y}||^3)] \tag{18}$$

$$= \frac{1}{2}\mathbb{E}_{t,\mathbf{x}_t}[(\Delta\mathbf{y})^T \frac{\partial^2 d}{\partial \mathbf{y}^2}\Delta\mathbf{y} + O(||\Delta\mathbf{y}||^3)] \tag{19}$$

, where we define $\Delta\mathbf{y} = \mathbf{f}_{\boldsymbol{\theta}^-}(\mathbf{x}_{t-\Delta_t}, t - \Delta_t) - \mathbf{f}_{\boldsymbol{\theta}}(\mathbf{x}_t, t)$.

Let's take the derivative with respect to $\boldsymbol{\theta}$:

$$\frac{1}{2}\frac{\partial}{\partial \boldsymbol{\theta}}\mathbb{E}_{t,\mathbf{x}_t}[(\mathbf{f}_{\boldsymbol{\theta}^-}(\mathbf{x}_{t-\Delta_t}, t - \Delta_t) - \mathbf{f}_{\boldsymbol{\theta}}(\mathbf{x}_t, t))^T \frac{\partial^2 d}{\partial \mathbf{y}^2}(\mathbf{f}_{\boldsymbol{\theta}^-}(\mathbf{x}_{t-\Delta_t}, t - \Delta_t) - \mathbf{f}_{\boldsymbol{\theta}}(\mathbf{x}_t, t)) + O(||\Delta\mathbf{y}||^3)] \tag{20}$$

$$= \mathbb{E}_{t,\mathbf{x}_t}[\frac{\partial^2 d}{\partial \mathbf{y}^2}(\mathbf{f}_{\boldsymbol{\theta}^-}(\mathbf{x}_{t-\Delta_t}, t - \Delta_t) - \mathbf{f}_{\boldsymbol{\theta}}(\mathbf{x}_t, t))\frac{\partial \mathbf{f}_{\boldsymbol{\theta}}}{\partial \boldsymbol{\theta}} + O(||\Delta\mathbf{y}||^3)]. \tag{21}$$

As

$$\mathbf{f}_{\boldsymbol{\theta}^-}(\mathbf{x}_{t-\Delta_t}, t - \Delta_t) = \mathbf{f}_{\boldsymbol{\theta}^-}(\mathbf{x}_t, t) - \frac{\partial \mathbf{f}_{\boldsymbol{\theta}^-}}{\partial \mathbf{x}_t}\frac{\partial \mathbf{x}_t}{\partial t}\Delta_t - \frac{\partial \mathbf{f}_{\boldsymbol{\theta}^-}}{\partial t}\Delta_t + O(\Delta_t^2), \tag{22}$$

we have

$$\mathbf{f}_{\boldsymbol{\theta}^-}(\mathbf{x}_{t-\Delta_t}, t - \Delta_t) - \mathbf{f}_{\boldsymbol{\theta}^-}(\mathbf{x}_t, t) = -(\frac{\partial \mathbf{f}_{\boldsymbol{\theta}^-}}{\partial \mathbf{x}_t}\frac{\partial \mathbf{x}_t}{\partial t} + \frac{\partial \mathbf{f}_{\boldsymbol{\theta}^-}}{\partial t})\Delta_t + O(\Delta_t^2). \tag{23}$$

Since $\mathbf{f}_{\boldsymbol{\theta}}(\mathbf{x}_t, t)$ has the same value as $\mathbf{f}_{\boldsymbol{\theta}^-}(\mathbf{x}_t, t)$, we can plug this into Eq. (21):

$$\mathbb{E}_{t,\mathbf{x}_t}[\frac{\partial^2 d}{\partial \mathbf{y}^2}(\mathbf{f}_{\boldsymbol{\theta}^-}(\mathbf{x}_{t-\Delta_t}, t - \Delta_t) - \mathbf{f}_{\boldsymbol{\theta}}(\mathbf{x}_t, t))\frac{\partial \mathbf{f}_{\boldsymbol{\theta}}}{\partial \boldsymbol{\theta}} + O(||\Delta\mathbf{y}||^3)] \tag{24}$$

$$= -\mathbb{E}_{t,\mathbf{x}_t}[\frac{\partial^2 d}{\partial \mathbf{y}^2}(\frac{\partial \mathbf{f}_{\boldsymbol{\theta}^-}}{\partial \mathbf{x}_t}\frac{\partial \mathbf{x}_t}{\partial t} + \frac{\partial \mathbf{f}_{\boldsymbol{\theta}^-}}{\partial t})\frac{\partial \mathbf{f}_{\boldsymbol{\theta}}}{\partial \boldsymbol{\theta}}\Delta_t + O(\Delta_t^2) + O(||\Delta\mathbf{y}||^3)] \tag{25}$$

$$= -\mathbb{E}_{t,\mathbf{x}_t}[\frac{\partial^2 d}{\partial \mathbf{y}^2}(\frac{\partial \mathbf{f}_{\boldsymbol{\theta}^-}}{\partial \mathbf{x}_t}\frac{\partial \mathbf{x}_t}{\partial t} + \frac{\partial \mathbf{f}_{\boldsymbol{\theta}^-}}{\partial t})\frac{\partial \mathbf{f}_{\boldsymbol{\theta}}}{\partial \boldsymbol{\theta}}\Delta_t + O(\Delta_t^2)]. \tag{26}$$

As the gradient is $O(\Delta_t)$, it becomes zero when $\Delta_t \to 0$, so it cannot be used for training. To make the gradient non-zero, Song et al. (2023) divide the by $\Delta_t$. Then, we have

$$\frac{\partial}{\partial \boldsymbol{\theta}} \mathcal{L}_{\text{CD}}(\boldsymbol{\theta}, \boldsymbol{\theta}^-) = \frac{\partial}{\partial \boldsymbol{\theta}} \mathbb{E}_{t, \mathbf{x}_t}[\frac{1}{\Delta_t} d(\mathbf{f}_{\boldsymbol{\theta}}(\mathbf{x}_t, t), \mathbf{f}_{\boldsymbol{\theta}^-}(\mathbf{x}_{t-\Delta_t}, t - \Delta_t))] \tag{27}$$

$$= -\mathbb{E}_{t, \mathbf{x}_t}[\frac{\partial^2 d}{\partial \mathbf{y}^2}(\frac{\partial \mathbf{f}_{\boldsymbol{\theta}^-}}{\partial \mathbf{x}_t} \frac{\partial \mathbf{x}_t}{\partial t} + \frac{\partial \mathbf{f}_{\boldsymbol{\theta}^-}}{\partial t})\frac{\partial \mathbf{f}_{\boldsymbol{\theta}}}{\partial \boldsymbol{\theta}} + O(\Delta_t)] \tag{28}$$

$$= -\mathbb{E}_{t, \mathbf{x}_t}[\frac{\partial^2 d}{\partial \mathbf{y}^2}(\frac{\partial \mathbf{f}_{\boldsymbol{\theta}^-}}{\partial \mathbf{x}_t} \frac{\partial \mathbf{x}_t}{\partial t} + \frac{\partial \mathbf{f}_{\boldsymbol{\theta}^-}}{\partial t})\frac{\partial \mathbf{f}_{\boldsymbol{\theta}}}{\partial \boldsymbol{\theta}}] \tag{29}$$

$$= -\mathbb{E}_{t, \mathbf{x}_t}[\frac{\partial^2 d}{\partial \mathbf{y}^2}(\frac{\partial \mathbf{f}_{\boldsymbol{\theta}^-}}{\partial \mathbf{x}_t}(-t \cdot \mathbf{s}_t(\mathbf{x}_t)) + \frac{\partial \mathbf{f}_{\boldsymbol{\theta}^-}}{\partial t})\frac{\partial \mathbf{f}_{\boldsymbol{\theta}}}{\partial \boldsymbol{\theta}}] \tag{30}$$

$$= \mathbb{E}_{t, \mathbf{x}_t}[\frac{\partial^2 d}{\partial \mathbf{y}^2}(\frac{\partial \mathbf{f}_{\boldsymbol{\theta}^-}}{\partial \mathbf{x}_t}(t \cdot \mathbf{s}_t(\mathbf{x}_t)) - \frac{\partial \mathbf{f}_{\boldsymbol{\theta}^-}}{\partial t})\frac{\partial \mathbf{f}_{\boldsymbol{\theta}}}{\partial \boldsymbol{\theta}}] \tag{31}$$

in the limit of $\Delta_t \to 0$.

**Hessian of $d$.** Here, we provide the Hessians of the L2 squared loss and the Pseudo-Huber loss.

1. If $d(\mathbf{x}, \mathbf{y}) = ||\mathbf{x} - \mathbf{y}||_2^2$, $\frac{\partial^2 d}{\partial \mathbf{y}^2}|_{\mathbf{y}=\mathbf{x}} = 2\mathbf{I}$.

2. If $d(\mathbf{x}, \mathbf{y}) = \sqrt{||\mathbf{x} - \mathbf{y}||_2^2 + c^2} - c$, $\frac{\partial^2 d}{\partial \mathbf{y}^2}|_{\mathbf{y}=\mathbf{x}} = \frac{1}{c}\mathbf{I}$.

### D.3 CONSISTENCY TRAINING

Song et al. (2023) show that Eq. (31) can be estimated without a pre-trained score network. From Tweedie's formula, we express the score function as

$$\mathbf{s}_t(\mathbf{x}_t) = \frac{\mathbb{E}_{p(\mathbf{x}|\mathbf{x}_t)}[\mathbf{x}] - \mathbf{x}_t}{t^2}. \tag{32}$$

Plugging this into Eq. (31), we have

$$\mathbb{E}_{t, \mathbf{x}_t}[\frac{\partial^2 d}{\partial \mathbf{y}^2}(\frac{\partial \mathbf{f}_{\boldsymbol{\theta}^-}}{\partial \mathbf{x}_t}(t \cdot \mathbf{s}_t(\mathbf{x}_t)) - \frac{\partial \mathbf{f}_{\boldsymbol{\theta}^-}}{\partial t})\frac{\partial \mathbf{f}_{\boldsymbol{\theta}}}{\partial \boldsymbol{\theta}}] = \mathbb{E}_{t, \mathbf{x}_t}[\frac{\partial^2 d}{\partial \mathbf{y}^2}(\frac{\partial \mathbf{f}_{\boldsymbol{\theta}^-}}{\partial \mathbf{x}_t}\frac{\mathbb{E}_{p(\mathbf{x}|\mathbf{x}_t)-\mathbf{x}_t}[\mathbf{x}]}{t} - \frac{\partial \mathbf{f}_{\boldsymbol{\theta}^-}}{\partial t})\frac{\partial \mathbf{f}_{\boldsymbol{\theta}}}{\partial \boldsymbol{\theta}}] \tag{33}$$

$$= \mathbb{E}_{t, \mathbf{x}_t}[\mathbb{E}_{p(\mathbf{x}|\mathbf{x}_t)}[\frac{\partial^2 d}{\partial \mathbf{y}^2}(\frac{\partial \mathbf{f}_{\boldsymbol{\theta}^-}}{\partial \mathbf{x}_t}\frac{\mathbf{x} - \mathbf{x}_t}{t} - \frac{\partial \mathbf{f}_{\boldsymbol{\theta}^-}}{\partial t})\frac{\partial \mathbf{f}_{\boldsymbol{\theta}}}{\partial \boldsymbol{\theta}}]] \tag{34}$$

$$= \mathbb{E}_{t, \mathbf{x}, \mathbf{x}_t}[\frac{\partial^2 d}{\partial \mathbf{y}^2}(\frac{\partial \mathbf{f}_{\boldsymbol{\theta}^-}}{\partial \mathbf{x}_t}\frac{\mathbf{x} - \mathbf{x}_t}{t} - \frac{\partial \mathbf{f}_{\boldsymbol{\theta}^-}}{\partial t})\frac{\partial \mathbf{f}_{\boldsymbol{\theta}}}{\partial \boldsymbol{\theta}}], \tag{35}$$

where we now have the expectation over three random variables $t, \mathbf{x}, \mathbf{x}_t$ and do not require a score function. In the next section, we will reverse-engineer an objective such that its gradient matches Eq. (35).

#### D.3.1 OBJECTIVE

It turns out that the following objective is the one we are looking for:

$$\mathcal{L}_{\text{CT}}(\boldsymbol{\theta}, \boldsymbol{\theta}^-) = \mathbb{E}_{t, \mathbf{x}, \boldsymbol{\epsilon}}[\frac{1}{\Delta_t} d(\mathbf{f}_{\boldsymbol{\theta}}(\mathbf{x} + t\boldsymbol{\epsilon}, t), \mathbf{f}_{\boldsymbol{\theta}^-}(\mathbf{x} + (t - \Delta_t)\boldsymbol{\epsilon}, t - \Delta_t))], \tag{36}$$

where $\boldsymbol{\epsilon} \sim \mathcal{N}(0, \mathbf{I})$ is a random noise vector. The objective in Eq. (36) is called the consistency training objective. We can show that the gradient of $\mathcal{L}_{\text{CT}}$ indeed matches Eq. (35) in the limit of

$\Delta_t \to 0$. First, we apply the Taylor expansion to the unweighted loss in Eq. (36):

$$\mathbb{E}_{t,\mathbf{x},\boldsymbol{\epsilon}}[d(\mathbf{f}_{\boldsymbol{\theta}}(\mathbf{x} + t\boldsymbol{\epsilon}, t), \underbrace{\mathbf{f}_{\boldsymbol{\theta}}(\mathbf{x} + t\boldsymbol{\epsilon}, t)}_{\mathbf{y}} + \underbrace{\mathbf{f}_{\boldsymbol{\theta}^-}(\mathbf{x} + (t - \Delta_t)\boldsymbol{\epsilon}, t - \Delta_t) - \mathbf{f}_{\boldsymbol{\theta}}(\mathbf{x} + t\boldsymbol{\epsilon}, t)}_{\Delta\mathbf{y}})] \tag{37}$$

$$= \mathbb{E}_{t,\mathbf{x},\boldsymbol{\epsilon}}[d(\mathbf{f}_{\boldsymbol{\theta}}(\mathbf{x} + t\boldsymbol{\epsilon}, t), \mathbf{f}_{\boldsymbol{\theta}}(\mathbf{x} + t\boldsymbol{\epsilon}, t)) + \frac{\partial d}{\partial \mathbf{y}}\Delta\mathbf{y} + (\Delta\mathbf{y})^T \frac{\partial^2 d}{\partial \mathbf{y}^2}\Delta\mathbf{y} + O(||\Delta\mathbf{y}||^3)] \tag{38}$$

$$= \mathbb{E}_{t,\mathbf{x},\boldsymbol{\epsilon}}[(\Delta\mathbf{y})^T \frac{\partial^2 d}{\partial \mathbf{y}^2}\Delta\mathbf{y} + O(||\Delta\mathbf{y}||^3)] \tag{39}$$

where we define $\Delta\mathbf{y}$ as $\Delta\mathbf{y} = \mathbf{f}_{\boldsymbol{\theta}^-}(\mathbf{x} + (t - \Delta_t)\boldsymbol{\epsilon}, t - \Delta_t) - \mathbf{f}_{\boldsymbol{\theta}}(\mathbf{x} + t\boldsymbol{\epsilon}, t)$.

Let's take the derivative with respect to $\boldsymbol{\theta}$:

$$\frac{\partial}{\partial \boldsymbol{\theta}}\mathbb{E}_t[\mathcal{L}_{\mathrm{CT}}(\boldsymbol{\theta}, \boldsymbol{\theta}^-)] = \mathbb{E}_{t,\mathbf{x},\boldsymbol{\epsilon}}[\frac{\partial^2 d}{\partial \mathbf{y}^2}(\mathbf{f}_{\boldsymbol{\theta}^-}(\mathbf{x} + (t - \Delta_t)\boldsymbol{\epsilon}, t - \Delta_t) - \mathbf{f}_{\boldsymbol{\theta}}(\mathbf{x} + t\boldsymbol{\epsilon}, t))\frac{\partial \mathbf{f}_{\boldsymbol{\theta}}}{\partial \boldsymbol{\theta}} + O(||\Delta\mathbf{y}||^3)]. \tag{40}$$

Using the Taylor expansion, we have

$$\mathbf{f}_{\boldsymbol{\theta}^-}(\mathbf{x} + (t - \Delta_t)\boldsymbol{\epsilon}, t - \Delta_t) = \mathbf{f}_{\boldsymbol{\theta}^-}(\mathbf{x} + t\boldsymbol{\epsilon}, t) - \frac{\partial \mathbf{f}_{\boldsymbol{\theta}^-}}{\partial \mathbf{x}}\boldsymbol{\epsilon}\Delta_t - \frac{\partial \mathbf{f}_{\boldsymbol{\theta}^-}}{\partial t}\Delta_t + O(\Delta_t^2), \tag{41}$$

$$\mathbf{f}_{\boldsymbol{\theta}^-}(\mathbf{x} + (t - \Delta_t)\boldsymbol{\epsilon}, t - \Delta_t) - \mathbf{f}_{\boldsymbol{\theta}^-}(\mathbf{x} + t\boldsymbol{\epsilon}, t) = -\frac{\partial \mathbf{f}_{\boldsymbol{\theta}^-}}{\partial \mathbf{x}}\boldsymbol{\epsilon}\Delta_t - \frac{\partial \mathbf{f}_{\boldsymbol{\theta}^-}}{\partial t}\Delta_t + O(\Delta_t^2). \tag{42}$$

Since $\mathbf{f}_{\boldsymbol{\theta}}(\mathbf{x} + t\boldsymbol{\epsilon}, t)$ has the same value as $\mathbf{f}_{\boldsymbol{\theta}^-}(\mathbf{x} + t\boldsymbol{\epsilon}, t)$, we can plug this into Eq. (40):

$$\mathbb{E}_{t,\mathbf{x},\boldsymbol{\epsilon}}[\frac{\partial^2 d}{\partial \mathbf{y}^2}(\mathbf{f}_{\boldsymbol{\theta}^-}(\mathbf{x} + (t - \Delta_t)\boldsymbol{\epsilon}, t - \Delta_t) - \mathbf{f}_{\boldsymbol{\theta}}(\mathbf{x} + t\boldsymbol{\epsilon}, t))\frac{\partial \mathbf{f}_{\boldsymbol{\theta}}}{\partial \boldsymbol{\theta}} + O(||\Delta\mathbf{y}||^3)] \tag{43}$$

$$= \mathbb{E}_{t,\mathbf{x},\boldsymbol{\epsilon}}[\frac{\partial^2 d}{\partial \mathbf{y}^2}(-\frac{\partial \mathbf{f}_{\boldsymbol{\theta}^-}}{\partial \mathbf{x}}\boldsymbol{\epsilon} - \frac{\partial \mathbf{f}_{\boldsymbol{\theta}^-}}{\partial t})\frac{\partial \mathbf{f}_{\boldsymbol{\theta}}}{\partial \boldsymbol{\theta}}\Delta_t + O(\Delta_t^2) \tag{44}$$

$$= \mathbb{E}_{t,\mathbf{x},\mathbf{x}_t}[\frac{\partial^2 d}{\partial \mathbf{y}^2}(-\frac{\partial \mathbf{f}_{\boldsymbol{\theta}^-}}{\partial \mathbf{x}}\frac{\mathbf{x}_t - \mathbf{x}}{t} - \frac{\partial \mathbf{f}_{\boldsymbol{\theta}^-}}{\partial t})\frac{\partial \mathbf{f}_{\boldsymbol{\theta}}}{\partial \boldsymbol{\theta}}\Delta_t + O(\Delta_t^2), \tag{45}$$

where in Eq. (45), we use the reparametrization trick $\mathbf{x} + t\boldsymbol{\epsilon} = \mathbf{x}_t$ and $\boldsymbol{\epsilon} = \frac{\mathbf{x}_t - \mathbf{x}}{t}$. Finally, we can show that Eq. (45) matches Eq. (35) in the limit of $\Delta_t \to 0$ and after dividing by $\Delta_t$:

$$\lim_{\Delta_t \to 0}\frac{\partial}{\partial \boldsymbol{\theta}}\mathcal{L}_{\mathrm{CD}}(\boldsymbol{\theta}, \boldsymbol{\theta}^-) = \lim_{\Delta_t \to 0}\frac{\partial}{\partial \boldsymbol{\theta}}\mathcal{L}_{\mathrm{CT}}(\boldsymbol{\theta}, \boldsymbol{\theta}^-) \tag{46}$$

We can add a weighting function $\omega(t)$ without affecting this equality, leading to the objective in Eq. (4).

## E   TRAINING OBJECTIVE OF TCMS

By substituting Eq. (7) into Eq. (6), we have:

$$\mathcal{L}_{\mathrm{CT}}(\mathbf{f}_{\boldsymbol{\theta},\boldsymbol{\theta}_0}^{\mathrm{trunc}}, \mathbf{f}_{\boldsymbol{\theta}^-,\boldsymbol{\theta}_0}^{\mathrm{trunc}}) = \lambda_b \frac{\omega(t')}{\Delta_{t'}}d(\mathbf{f}_{\boldsymbol{\theta}}(\mathbf{x} + t'\boldsymbol{\epsilon}, t'), \mathbf{f}_{\boldsymbol{\theta}_0^-}(\mathbf{x} + (t' - \Delta_{t'})\boldsymbol{\epsilon}, t' - \Delta_{t'}) \tag{47}$$

$$+ (1 - \lambda_b)\int_{t \in S_{t'}} \bar{\psi}_t(t)\frac{\omega(t)}{\Delta_t}d(\mathbf{f}_{\boldsymbol{\theta}}(\mathbf{x} + t\boldsymbol{\epsilon}, t), \mathbf{f}_{\boldsymbol{\theta}_0^-}(\mathbf{x} + (t - \Delta_t)\boldsymbol{\epsilon}, t - \Delta_t)dt \tag{48}$$

$$+ (1 - \lambda_b)\int_{t \in S_{t'}^-} \bar{\psi}_t(t)\frac{\omega(t)}{\Delta_t}d(\mathbf{f}_{\boldsymbol{\theta}}(\mathbf{x} + t\boldsymbol{\epsilon}, t), \mathbf{f}_{\boldsymbol{\theta}^-}(\mathbf{x} + (t - \Delta_t)\boldsymbol{\epsilon}, t - \Delta_t)dt \tag{49}$$

The first two terms in RHS represent the boundary loss, and the last term is the consistency loss. Let us define $\Delta_t = (1 + 8 \cdot \mathrm{sigmoid}(-t))(1 - r)t$. We define $t_m$ to be the smallest point such that $t_m - \Delta_{t_m} = t'$. Then the volume of the set $S_{t'} := \{t \in \mathbb{R} : t' \leq t \leq t' + \Delta_t\}$ is $\Delta_{t_m}$. We assume

$\bar{\psi}_t$ is properly designed to be upper bounded by a finite value (see Sec. 4.4). In the limit of $r \to 1$, we simplify the the second term in the above:

$$(1 - \lambda_b) \int_{t \in S_{t'}} \bar{\psi}_t(t) \frac{\omega(t)}{\Delta_t} d(\mathbf{f}_{\boldsymbol{\theta}}(\mathbf{x} + t\boldsymbol{\epsilon}, t), \mathbf{f}_{\boldsymbol{\theta}_0^-}(\mathbf{x} + (t - \Delta_t)\boldsymbol{\epsilon}, t - \Delta_t) dt \tag{50}$$

$$= (1 - \lambda_b) \int_{t \in S_{t'}} \bar{\psi}_t(t') \frac{\omega(t')}{\Delta_{t'}} d(\mathbf{f}_{\boldsymbol{\theta}}(\mathbf{x} + t'\boldsymbol{\epsilon}, t'), \mathbf{f}_{\boldsymbol{\theta}_0^-}(\mathbf{x} + (t' - \Delta_{t'})\boldsymbol{\epsilon}, t' - \Delta_{t'}) + O(1) dt \tag{51}$$

$$= (1 - \lambda_b) \bar{\psi}_t(t') \frac{\omega(t')}{\Delta_{t'}} d(\mathbf{f}_{\boldsymbol{\theta}}(\mathbf{x} + t'\boldsymbol{\epsilon}, t'), \mathbf{f}_{\boldsymbol{\theta}_0^-}(\mathbf{x} + (t' - \Delta_{t'})\boldsymbol{\epsilon}, t' - \Delta_{t'}) Vol(S_{t'}) + \int_{t \in S_{t'}} O(1) dt \tag{52}$$

$$\approx (1 - \lambda_b) \bar{\psi}_t(t') \frac{\omega(t')}{\Delta_{t'}} d(\mathbf{f}_{\boldsymbol{\theta}}(\mathbf{x} + t'\boldsymbol{\epsilon}, t'), \mathbf{f}_{\boldsymbol{\theta}_0^-}(\mathbf{x} + (t' - \Delta_{t'})\boldsymbol{\epsilon}, t' - \Delta_{t'}) \Delta_{t'} + \int_{t \in S_{t'}} O(1) dt \tag{53}$$

$$\approx (1 - \lambda_b) \bar{\psi}_t(t') \omega(t') d(\mathbf{f}_{\boldsymbol{\theta}}(\mathbf{x} + t'\boldsymbol{\epsilon}, t'), \mathbf{f}_{\boldsymbol{\theta}_0^-}(\mathbf{x} + (t' - \Delta_{t'})\boldsymbol{\epsilon}, t' - \Delta_{t'}), \tag{54}$$

where in Eq. (51) we apply the Taylor expansion to the integrand, and in Eq. (53), we can see that $\Delta_{t_m} = (1 + 8 \cdot \text{sigmoid}(-t_m))(1 - r)t_m$ goes to zero as $r \to 1$. Thus, $t_m \to t'$ and then $Vol(S_{t'})/\Delta_{t'} = \Delta_{t_m}/\Delta_{t'} = \frac{(1+8\cdot\text{sigmoid}(-t_m))t_m(1-r)}{(1+8\cdot\text{sigmoid}(-t'))t'(1-r)} = 1$ in the limit. Hence, the boundary loss is

$$(\lambda_b \frac{\omega(t')}{\Delta_{t'}} + (1 - \lambda_b) \bar{\psi}_t(t') \omega(t')) d(\mathbf{f}_{\boldsymbol{\theta}}(\mathbf{x} + t'\boldsymbol{\epsilon}, t'), \mathbf{f}_{\boldsymbol{\theta}_0^-}(\mathbf{x} + (t' - \Delta_{t'})\boldsymbol{\epsilon}, t' - \Delta_{t'}) \tag{55}$$

$$\approx \lambda_b \frac{\omega(t')}{\Delta_{t'}} d(\mathbf{f}_{\boldsymbol{\theta}}(\mathbf{x} + t'\boldsymbol{\epsilon}, t'), \mathbf{f}_{\boldsymbol{\theta}_0^-}(\mathbf{x} + (t' - \Delta_{t'})\boldsymbol{\epsilon}, t' - \Delta_{t'}), \tag{56}$$

where the first term ($O(1/\Delta_t)$) dominates the second term ($O(1)$). We hence arrive at Eq. (9):

$$\mathcal{L}_{\text{CT}}(\mathbf{f}_{\boldsymbol{\theta}, \boldsymbol{\theta}_0^-}^{\text{trunc}}, \mathbf{f}_{\boldsymbol{\theta}^-, \boldsymbol{\theta}_0^-}^{\text{trunc}}) \approx \lambda_b \underbrace{\frac{\omega(t')}{\Delta_{t'}} d(\mathbf{f}_{\boldsymbol{\theta}}(\mathbf{x} + t'\boldsymbol{\epsilon}, t'), \mathbf{f}_{\boldsymbol{\theta}_0^-}(\mathbf{x} + (t' - \Delta_{t'})\boldsymbol{\epsilon}, t' - \Delta_{t'}))}_{\text{Boundary loss}:=\mathcal{L}_B(\mathbf{f}_{\boldsymbol{\theta}}, \mathbf{f}_{\boldsymbol{\theta}_0^-})} \tag{57}$$

$$+ (1 - \lambda_b) \underbrace{\mathbb{E}_{\bar{\psi}_t}[\frac{\omega(t)}{\Delta_t} d(\mathbf{f}_{\boldsymbol{\theta}}(\mathbf{x} + t\boldsymbol{\epsilon}, t), \mathbf{f}_{\boldsymbol{\theta}^-}(\mathbf{x} + (t - \Delta_t)\boldsymbol{\epsilon}, t - \Delta_t)]}_{\text{Consistency loss}:=\mathcal{L}_C(\mathbf{f}_{\boldsymbol{\theta}}, \mathbf{f}_{\boldsymbol{\theta}^-})}, \tag{58}$$

where because $\Delta_t \to 0$, we can approximate $S_{t'}^- \approx (t', T]$, leading to Eq. (58).

In practice, we set $r$ close to one during the truncated training stage (see Section F). Note that here, the boundary loss can dominate the consistency loss when $\mathbf{f}_{\boldsymbol{\theta}}$ and $\mathbf{f}_{\boldsymbol{\theta}_0^-}$ are sufficiently different around $t = t'$. However, in practice, as we set $\Delta_t$ to be small enough but not all the way to zero, and as we use Pseudo-Huber loss (see Section F) with small $c$ value that normalizes the effect of the loss magnitude on the gradient norm, we can balance the training.

## F  IMPLEMENTATION DETAILS

We provide detailed information about our implementation in the following.

**Model initialization and architecture:**  All stage 1 models are initialized from pre-trained EDM or EDM2 checkpoints as suggested by Geng et al. (2024). For CIFAR-10, we use EDM's DDPM++ architecture, which is slightly smaller than iCT's NCSN++. For ImageNet $64 \times 64$, we employ EDM2-S (280M parameters) and EDM2-XL (approximately 1.1B parameters) architectures. EDM2-S is slightly smaller than iCT's ADM architecture (296M parameters).

**Model parameterization:** Following EDM, we parameterize consistency models $\mathbf{f}_{\boldsymbol{\theta}}$ as $\mathbf{f}_{\boldsymbol{\theta}} = c_{\text{out}}(t)\mathbf{F}_{\boldsymbol{\theta}}(\mathbf{x}, t) + c_{\text{skip}}(t)\mathbf{x}$, where $c_{\text{out}}(t) = \frac{t\sigma_{\text{data}}}{\sqrt{\sigma_{\text{data}}^2 + t^2}}$, $c_{\text{skip}}(t) = \frac{\sigma_{\text{data}}^2}{\sigma_{\text{data}}^2 + t^2}$, and $\sigma_{\text{data}} = 0.5$.

**Training details:** We set $\Delta_t = (1 + 8 \cdot \text{sigmoid}(-t))(1 - r)t$, where $r = \max\{1 - 1/2^{\lceil i/25000 \rceil}, 0.999\}$ for CIFAR-10 and $\max\{1 - 1/4^{\lceil i/25000 \rceil}, 0.9961\}$ for ImageNet $64 \times 64$, with $i$ being the training iteration. For CIFAR-10, we train for 250K iterations in Stage 1 and 200K iterations in Stage 2. For ImageNet $64 \times 64$, EDM2-S is trained for 150K iterations in Stage 1 and 120K iterations in Stage 2, while EDM2-XL is trained for 40K iterations in Stage 2 only. See Fig. 1(b) for the FID evolution during training. For the second stage, we start with the maximum $r$ values (i.e., 0.999 or 0.9961) and do not change them. The weighting function $\omega(t)$ is set to 1 for CIFAR-10 and $\Delta_t/c_{\text{out}}(t)^2$ for ImageNet $64 \times 64$. As suggested by Song & Dhariwal (2023); Geng et al. (2024), we use the Pseudo-Huber loss function $d(\mathbf{x}, \mathbf{y}) = \sqrt{||\mathbf{x} - \mathbf{y}||_2^2 + c^2} - c$, with $c = 1e - 8$ for CIFAR-10 and $c = 0.06$ for ImageNet $64 \times 64$. This is especially crucial for our method as the boundary loss can dominate the consistency loss. The boundary loss compares the outputs from the different model $\mathbf{f}_{\boldsymbol{\theta}}$ and $\mathbf{f}_{\boldsymbol{\theta}_0}$, it tends to be larger than the consistency loss, but Pseudo-Huber loss effectively normalize the effect of the loss magnitude on the gradient norm. For ImageNet $64 \times 64$, we employ mixed-precision training with dynamic loss scaling and use power function EMA (Karras et al., 2024) with $\gamma = 6.94$ (without post-hoc EMA search).

**Learning rate schedules:** EDM2 (Karras et al., 2024) architectures require a manual decay of the learning rate. Karras et al. (2024) suggest using the inverse square root schedule $\frac{\alpha_{\text{ref}}}{\sqrt{\max(t/t_{\text{ref}}, 1)}}$. For the first stage training of EDM2-S on ImageNet, we use $t_{\text{ref}} = 2000$ and $\alpha_{\text{ref}} = 1e - 3$ following Geng et al. (2024). For the second stage training of EDM2-S, we use $t_{\text{ref}} = 8000$ and $\alpha_{\text{ref}} = 5e - 4$.

Second stage training of EDM2-XL is initialized with the ECM2-XL checkpoint from Geng et al. (2024). During the second stage, we use $t_{\text{ref}} = 8000$ and $\alpha_{\text{ref}} = 1e - 4$ for EDM2-XL.

**Time step sampling:** For the first stage training, we use a log-normal distribution for $\bar{\psi}_t$. For CIFAR-10, we use a mean of -1.1 and a standard deviation of 2.0 following Song & Dhariwal (2023). For ImageNet, we use a mean of -0.8 and a standard deviation of 1.6 following Geng et al. (2024).

For EDM2-XL, we also explore $t' = 1.5$ for truncated training, adjusting $\nu$ to 2 to ensure $\bar{p}_t$ has high probability mass around $t' = 1.5$ and also has a long tail as discussed in Sec. 4.4. This way, we get the FID of 2.15, which is slightly better than the result in Table 2.

During two-step generation, we evaluate the model at $t = 80, 1$ on CIFAR-10 and $t = 80, 1.526$ for ImageNet.

# G   UNCURATED GENERATED SAMPLES

We provide the uncurated generated samples in Fig. 9-11.

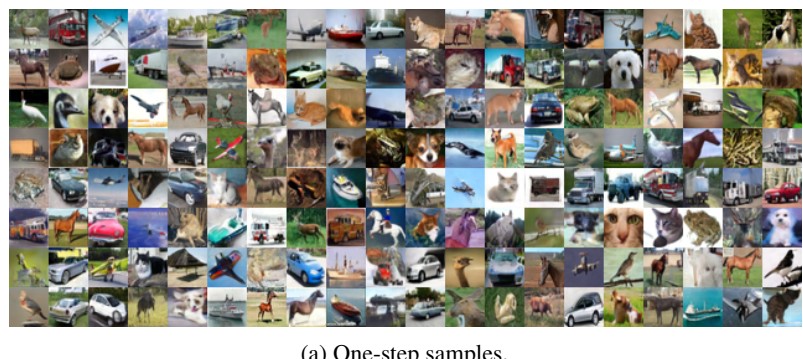

(a) One-step samples.

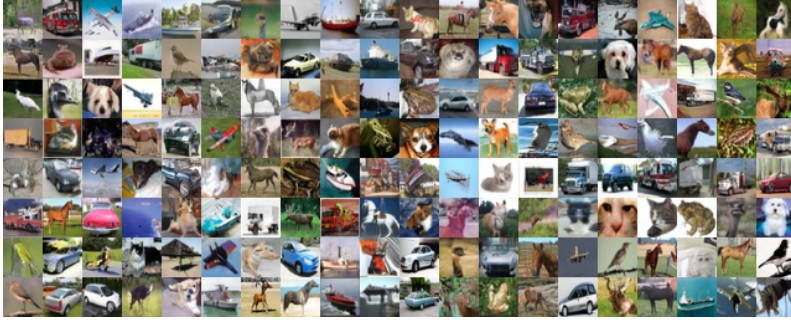

(b) Two-step samples.

Figure 9: Uncurated one-step and two-step samples on CIFAR-10.

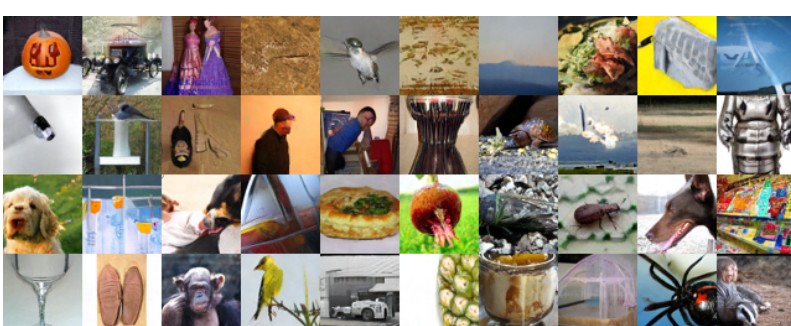

(a) One-step samples.

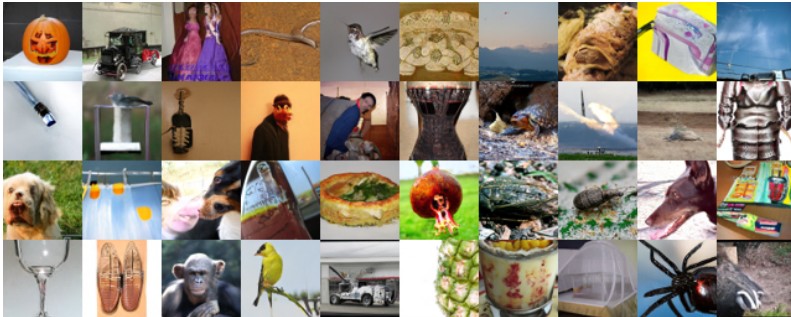

(b) Two-step samples.

Figure 10: Uncurated one-step and two-step samples on ImageNet (EDM2-S).

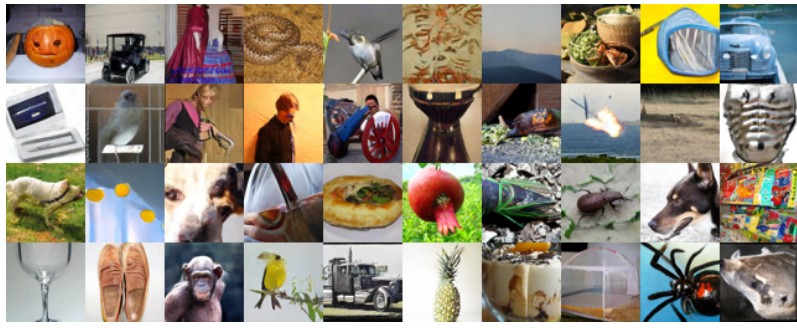

(a) One-step samples.

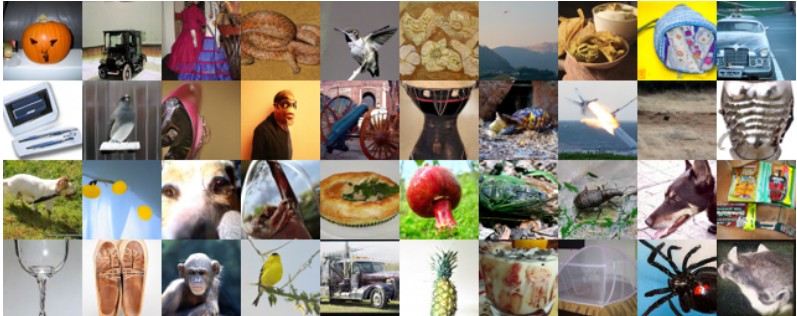

(b) Two-step samples.

Figure 11: Uncurated one-step and two-step samples on ImageNet (EDM2-XL).

