# OpenReview forum: "Truncated Consistency Models"
_ICLR.cc/2025/Conference — ICLR 2025 Poster_

### Official Review · Reviewer_NmLx · 2024-10-28

**Soundness:** 3
**Presentation:** 4
**Contribution:** 2
**Rating:** 8
**Confidence:** 4

**Summary:**

This paper identifies a shortcoming of consistency models: their joint temporal parameterization via a single network which limits their learnability. By noticing that early denoising timesteps, irrelevant for the final generation task, are neglected at the end of the training, the authors propose to train a consistency model on large noise levels only, with a pretrained consistency model serving as initial condition at the cutoff point. The relevance and the advantage of this approach are empirically demonstrated as the resulting model obtains state-of-the-art performance with limited number of parameters and reduced instability.

**Strengths:**

This paper is **very well written**: it is easy to read and understand. All ideas are presented and articulated clearly. The method's description and results are presented in a **simple and efficient manner**, making it easy to reimplement. Given its generality, I believe that Truncated Consistency Models may **become a widespread trick** to improve consistency models' performance.

The model is particularly **well motivated**, and its design **well supported** by sound experiments. The latter suscessfully highlight the denoising-generation tradeoff of standard consistency models. The efficiency of the proposed method to improve the generation performance at later training steps is demonstrated in the experiments. **Additional insights on design choices** (cutoff time, relevance of more training stages, etc.) are appreciated.

To my knowledge, both the presented insights and model are novel, even though the truncation principle has become a standard technique in diffusion (Balaji et al., 2022; Zheng et al., 2023 -- that could be cited in the related work) and consistency (Heek et al., 2024) models. Overall, the paper's contribution has a **good significance potential** as the presented insights are valuable for all practitioners, the experimental results are appealing and the method is generally applicable. The denoising-generation duality also echoes more fundamental work unveiling phase transition phenomena in diffusion models (Ambrogioni, 2023; Biroli et al., 2024; Sclocchi et al., 2024).

Ambrogioni. The statistical thermodynamics of generative diffusion models: Phase transitions, symmetry breaking and critical instability. arXiv:2310.17467, 2023.\
Balaji et al. eDiff-I: Text-to-Image Diffusion Models with Ensemble of Expert Denoisers. arXiv:2211.01324, 2022.\
Zheng et al. Truncated Diffusion Probabilistic Models and Diffusion-based Adversarial Auto-Encoders. ICLR 2023.\
Biroli et al. Dynamical Regimes of Diffusion Models. arXiv:2402.18491, 2024.\
Heek et al. Multistep Consistency Models. arXiv:2403.06807, 2024.\
Sclocchi et al. A Phase Transition in Diffusion Models Reveals the Hierarchical Nature of Data. arXiv:2402.16991, 2024.

**Weaknesses:**

While they are not critical and may not significantly change my assessment, I would like the authors to tackle the following weaknesses.

### Unnecessary decomposition of the loss

I do not understand the necessity of the loss decomposition in Eq. (6). To my understanding, it amounts to giving a specific sampling weight for timesteps close to $t'$ in $\psi$. The parameterization of Eq. (5) should then handle the corner case. Additionally, it appears that $S_{t'} = \{t'\}$, so the notations may be simplified in this part of the paper.

### Additional insights & reproducibility

The following discussions / content would benefit the paper.
- The code used by the authors for the presented experiments should be included in the supplementary material for further reproducibility.
- One can notice in Figure 3 that the denoising FID increases at the beginning of stage 2. I would be interested in the authors' opinion on this phenomenon.
- Still in Figure 3, can the authors explain why the denoising FIDs do not start from the same value between stage 1 and stage 2 (assuming they start from the same checkpoint)?
- While the authors acknowledge an additional training and memory cost in Section 6, I am not sure they include the additional cost induced by the supplemental training iterations to complement the pretrained consistency model (up to 200k additional optimization steps based on the different plots).

### Minor issues

- Section 4.2 insists on the forgetting by the truncated model on the lower noise levels. This is not an advantage or a confirmation of the model's abilities, but a byproduct of the truncated training loss.
- The number of optimization steps is not specified in the hyperparameters listed in Section D.
- In Figure 3, the timestep $t = 1$ should be placed in the second row to be consistent with the rest of the paper.
- The paper should specify where the baseline results from Tables 1 and 2 are reported from.

**Questions:**

I have no specific question -- cf. the above comments. Overall, I recommend an "accept" but am waiting for the authors' response to my comments and look forward to discussing with the other reviewers on the matter.

---

> ### Author Response · Authors · 2024-11-24
>
> We appreciate the reviewer’s valuable comments.
>
> ## The truncation principle has become a standard technique in diffusion (Balaji et al., 2022; Zheng et al., 2023 -- that could be cited in the related work) and consistency (Heek et al., 2024) models.
>
> Thank you for those references! We have added Balaji et al., 2022; Zheng et al., 2023 to our related work section. Note that since consistency models solve a more difficult task (learning to integrate PF-ODE) than diffusion models (learning the drift of PF-ODE), they can benefit more from such a strategy but also require a specific parameterization (Eq. 5) to satisfy the boundary condition.
>
> > Heek et al. Multistep Consistency Models.
>
> We clarify that our motivation and implementation of truncated training are distinct from Heek et al. Their models predict the noised data while our models always predict the clean data, which motivates us to design new parameterization (Eq. 5). As a result, the multistep consistency model is not able to do one-step generation while we exclusively aim for it.
>
> ## I do not understand the necessity of the loss decomposition in Eq. (6).
> > To my understanding, it amounts to giving a specific sampling weight for timesteps close to t'.
>
> Eq. (6) is a simple decomposition of the original loss Eq. (4), and we haven't introduced any specific weighting yet. We need this decomposition to motivate such a weighting strategy.
>
> ## Additionally, it appears that $S_{t'} = t'$, so the notations may be simplified in this part of the paper.
>
> The equality only holds in the limit of $\Delta t \to 0$. Since we assume discrete-time consistency training, the notations like $S_{t’}$ help clarify that the boundary loss vanishes only when $\Delta t$ is close to zero.
>
> ## The code used by the authors for the presented experiments should be included in the supplementary material for further reproducibility.
>
> We are still going through the internal release process, but we expect the code to be released soon.
>
> ## One can notice in Figure 3 that the denoising FID increases at the beginning of stage 2. I would be interested in the authors' opinion on this phenomenon.
>
> This behavior is intended and aligns well with our motivation--moving the capacity dedicated to denoising to generation. It shows the model effectively "forgets" the unnecessary denoising tasks.
>
> ## Still in Figure 3, can the authors explain why the denoising FIDs do not start from the same value between stage 1 and stage 2 (assuming they start from the same checkpoint)?
>
> Those two are from different settings. We have updated Fig. 3 to match their initial checkpoints. Thank you for pointing this out!
>
> ## While the authors acknowledge an additional training and memory cost in Section 6, I am not sure they include the additional cost induced by the supplemental training iterations to complement the pretrained consistency model (up to 200k additional optimization steps based on the different plots).
>
> Indeed, we have considered the per-iteration training cost only. We agree with the reviewer and have updated our limitation section accordingly. Thanks for catching this!
>
> ## Section 4.2 insists on the forgetting by the truncated model on the lower noise levels. This is not an advantage or a confirmation of the model's abilities, but a byproduct of the truncated training loss.
>
> This behavior confirms our motivation for the redistribution of the neural net capacity. That said, if it had not happened, it would have been difficult to justify the enhanced performance coming from the truncated training (if the model did not forget denoising, why would we expect the generation to be improved?).
>
> ## The number of optimization steps is not specified in the hyperparameters listed in Section D.
>
> The number of iterations is as follows:
> - CIFAR-10: Stage 1: 250K iterations, Stage 2: 200K iterations
> - ImageNet 64x64
>   - EDM2-S: Stage 1: 150K iterations, Stage 2: 120K iterations
>   - EDM2-XL: Stage 1: 100K iterations (we used the pre-trained checkpoint), Stage 2: 40K iterations
>
> The FID evolution on CIFAR-10 and ImageNet 64x64 (EDM2-S) during training is shown in Fig. 1(b).
> We have reflected this in the appendix.
>
> ## The paper should specify where the baseline results from Tables 1 and 2 are reported from.
>
> We mostly borrow the baseline results from the original papers. We have updated Sec. 4.3 to clarify this.

---

> > ### Comment · Reviewer_NmLx · 2024-11-24
> > **Acknowledgment**
> >
> > I would like to thank the authors for their response, which addressed my comments. I am looking forward to the release of the source code. I maintain my score for the time being and look forward to discussing this submission with the other reviewers.

---

### Official Review · Reviewer_2JKb · 2024-10-31

**Soundness:** 3
**Presentation:** 3
**Contribution:** 3
**Rating:** 6
**Confidence:** 3

**Summary:**

This paper proposes an interesting way to improve the training of consistency models. It points out, simutaneously learn to denoise and generate can require more capacity. Based on the belief, the paper proposes to separately use two models for denoising and generation, with a two-stage training procedure that first train a standard consistency model, and then train a generation model. The challenging part is the training loss and weighting function need to be carefully designed to enforce the boundary condition between two time ranges. The proposed approach seems to successfully train stably with continuously decreased FID, and improves upon existing work ECM.

**Strengths:**

- The paper largely focus on an orthogonal problem of network capacity, which can be possibly combined with many consistency distillation works.
- Train two separate models with a boundary condition to enforce consistency is empirically challenging (though theoretically straightforward), but the paper shows that it is possible. This might be informative for subsequent works that require different models across time ranges.

**Weaknesses:**

- While the proposed method improves considerably on small models and few time steps, the improvement on the highest quality 2-step EDM2-XL is negible. The applicability of the proposed method on large-scale models is questionable.
- The proposed method seems to still perform worse than SiD.

**Questions:**

Besides the weaknesses listed above, I also have some questions.
- I am not an expert on consistency models. But there are many other works on consistency distillation, such as CTM / TCD. Why not comparing against them?
- SiD seems to get better FID with significantly fewer parameters. Why variational score distillation methods are listed in separate categories and not compared against consistency models?
- How does the proposed method apply to guided sampling / text-to-image applications? How does guidance scale impact the proposed method? (Including more guided sampling results would improve the submission.)

---

> ### Author Response · Authors · 2024-11-24
>
> We appreciate the reviewer’s valuable comments.
>
> ## While the proposed method improves considerably on small models and few time steps, the improvement on the highest quality 2-step EDM2-XL is negligible. The applicability of the proposed method on large-scale models is questionable.
>
> EDM2-S model (with which our method achieves significant gain) has 280M parameters and is not small for the ImageNet 64x64 dataset (its size is similar to the widely used ADM architecture). Methods like variational score distillation use similar-sized networks and still can produce good FID on this dataset. The fact that consistency models require a larger network (1.1B) and more sampling steps makes them less scalable, and solving this issue is the main objective of our paper. Also, we experimented with the text-to-image setting and found that our method works well on a large-scale setting. See the general response above.
>
> Also, we contacted the ECT authors for more details in training ECT with EDM2-XL (which was not released at the time of the submission), but still could not reproduce its results (the training diverged), especially with a large batch size, which could be due to e.g. the training details for EDM2-XL that are not available to us. We thus ended up using the checkpoint provided by the authors. We believe that with the right configuration for large batch training, the 2-step FID of EDM2-XL could be improved further.
>
> It is in fact surprising that the truncated training works at all with EDM2-XL, given that the untruncated training with EDM2-XL does not even converge –a preliminary indication that the truncated training can be more stable.
>
>
>
> ## Why variational score distillation methods are listed in separate categories and not compared against consistency models?
>
> This is a good point! There are mainly two reasons: 1) Variational score distillation requires training a separate model (e.g. the fake score network) to estimate the gradient of their distribution-matching loss. This requires a GAN-like alternative training procedure, which introduces additional engineering burden and computational overhead. 2) The main purpose of this paper is to understand and improve the training of consistency models. We believe that our method is a step towards closing the gap between consistency models and variational score distillation in FID.
>
> ## SiD seems to get better FID with significantly fewer parameters.
>
> SiD is a variational score distillation method. It requires training a separate model and also a GAN-like alternative training procedure, which introduces more training instabilities and computational overhead. While the performance of SiD is indeed promising, we believe that the understanding of why it works is still lacking. For instance, SiD requires a very large batch size (8K) to make it not diverge on ImageNet 64x64. Moreover, their final loss function is an extrapolation of the two different loss functions they suggested, which is not justified by theory. On the contrary, consistency models do not use an alternative training procedure, require less computational overhead, and their loss function is supported by theory.
>
> ## But there are many other works on consistency distillation, such as CTM / TCD. Why not comparing against them?
>
> To our knowledge, our model outperforms all published consistency distillation/training methods on CIFAR-10 and ImageNet 64x64. We do compare with CTM (or TCD) in Table 1 where we outperform it by a large margin. Note that we have excluded the methods that rely on GAN loss, such as CTM + GAN, for fair comparison.
>
> ## How does the proposed method apply to guided sampling / text-to-image applications? How does guidance scale impact the proposed method? (Including more guided sampling results would improve the submission.)
>
> See the general response above.

---

> > ### Comment · Reviewer_2JKb · 2024-11-25
> > **Thanks for the rebuttal**
> >
> > I would like to thank the authors for the clarifications. I decide to retain my score.

---

### Official Review · Reviewer_jsgK · 2024-11-01

**Soundness:** 2
**Presentation:** 3
**Contribution:** 2
**Rating:** 6
**Confidence:** 3

**Summary:**

This work proposes a two-stage consistency training (CT) approach by dividing the time interval into two segments. Stage 1 employs standard CT, serving as a boundary anchor for Stage 2, which focuses on larger time consistency training. This method, the authors claim, emphasizes generation over denoising.

**Strengths:**

The paper is well-structured and clear. I appreciate the extensive experiments illustrating the trade-off between denoising and generation in consistency training. Additionally, the empirical results seem competitive to baselines.

**Weaknesses:**

1. The approach appears a bit over-engineered, with numerous handcrafted designs and hyperparameters, such as the weighting function $\psi_t(t)$, $\lambda_b$, $N_B$, $\Delta_t$, $\Delta_{t'}$, and the interval division $t'$.

2. How should one determine the terminate point of Stage 1 training? Monitoring its progress may introduce additional complexity. The number of training iterations for Stage 1 may also represent a crucial hyperparameter that may require further ablation studies.

3. The paper lacks ablation studies for certain hyperparameters relevant to the proposed method, such as $N_B$ (or $\rho$), $\Delta_t$, $\Delta_{t'}$.

**Questions:**

1. Several studies, such as [1, 2], have proposed truncating the time interval in diffusion model training to address issues like the unbounded score problem or high Lipschitz continuity when $t\approx0$. In principle, similar phenomena could arise in consistency training, as it aims to learn a noise-to-data mapping that may also exhibit large Lipschitz continuity. How does the proposed method relate to this existing literature?


2. How can one ensure that the trained CT model, using the proposed parameterization in Eq. (5), will ultimately be continuous at the splitting time $t'$? If one performs multi-step sampling and selects (some) sampling timesteps close to $t'$, this could potentially cause issues?



[1] Kim, D., Shin, S., Song, K., Kang, W., & Moon, I. C. (2021). Soft truncation: A universal training technique of score-based diffusion model for high precision score estimation.

[2] Yang, Z., Feng, R., Zhang, H., Shen, Y., Zhu, K., Huang, L., ... & Cheng, F. (2023). Eliminating lipschitz singularities in diffusion models.

---

> ### Author Response · Authors · 2024-11-24
>
> We appreciate the reviewer’s valuable comments.
>
> ## The approach appears a bit over-engineered, with numerous handcrafted designs and hyperparameters.
>
> We discuss the hyperparameters introduced by truncated training in the general response. Our method does introduce a few additional hyperparameters, but we believe this is not always bad as long as one of these conditions is met: 1) there is a good guiding principle to determine their value, 2) the hyperparameters can easily be transferred from the smaller to larger models/datasets, or 3) the results are stable with a wide range of values.
>
> Also, when developing our method, we intentionally avoided tuning these hyperparameters for any specific model or dataset and **used exactly the same values for all experiments to ensure the generalizability of our method.** Hence, the additional complexity of our framework is marginal.
>
> ## How should one determine the terminate point of Stage 1 training? Monitoring its progress may introduce additional complexity. The number of training iterations for Stage 1 may also represent a crucial hyperparameter that may require further ablation studies.
>
> For both stages 1 and 2, we simply monitor the 1-step FID during training and terminate the training when it converges. Determining when to terminate a generative model training can be done in many ways (e.g., validation loss or other performance metrics) and is not specific to our paper.
>
> ## The paper lacks ablation studies for certain hyperparameters relevant to the proposed method, such as $\rho, \Delta_t, \Delta_{t'}$.
>
> The effect of $\rho$ is shown in Fig. 4(c). $\Delta_t$ (or $\Delta_{t'}$) follows the prior work ECT without modification. It is a general design choice in all consistency models that is not particularly related to the idea of truncated training.
>
> ## Several studies, such as [1, 2], have proposed truncating the time interval in diffusion model training to address issues like the unbounded score problem or high Lipschitz continuity when $t \approx 0$. How does the proposed method relate to this existing literature?
>
> We thank the reviewer for pointing out these two related works. While these references and our work both employ truncation techniques, fundamental differences exist in both motivation and implementation. These works primarily focus on stabilizing regression targets during diffusion model training, whereas our approach addresses the capacity allocation in consistency models. As they focus on diffusion models, there is no notion of boundary condition in these two works, which, however, motivates a new parameterization in TCM.
>
> In those papers, the lower bound of the truncated time interval is close to 0 as they aim to avoid an unstable regression target around t=0. In our case, we need the lower bound of the truncated time interval to be sufficiently large to represent the point where the consistency mapping transitions from noise reduction (denoising) to sample generation.
>
> In fact, with the standard EDM pre-conditioning, the truncation proposed by these two papers is not necessary for stable training. However, in our case, the effect of the truncated training comes from the explicit redistribution of the capacity and cannot be achieved by simply reparameterizing the neural network (e.g., EDM pre-conditioning).
>
>
> ## How can one ensure that the trained CT model, using the proposed parameterization in Eq. (5), will ultimately be continuous at the splitting time $t'$? If one performs multi-step sampling and selects (some) sampling timesteps close to $t'$, this could potentially cause issues?
>
> We agree that, in our parameterization, there could be a discontinuity around t' as we use the stage-1 model when $t < t'$ and the stage-2 model when $t \geq t’$. However, since we only evaluate our stage-2 model on $t\geq t’$ (to ultimately discard the stage-1 model after training), it does not happen during generation. Empirically, we obtain the state-of-the-art 2-step FID by evaluating the model at $t=t'$ in our experiments.

---

> > ### Comment · Reviewer_jsgK · 2024-11-27
> > **Thank You for the Authors' Comments; Some Concerns Remain**
> >
> > Thank you for the authors' comments. However, my concerns remain partially unresolved:
> >
> > 1. While I acknowledge that introducing hyperparameters is not inherently problematic, my questions are still not fully addressed. Specifically:
> >    - The paper lacks ablation studies for certain hyperparameters (refer to W3).
> >    - Guidance on determining when to terminate Stage 1 training is missing. For instance, what level of degradation occurs if Stage 1 training is imperfect? Does this significantly impact one-step or two-step sampling?
> >
> > 2. Regarding [1, 2], I believe the fundamental issue is similar: addressing the challenge of large Lipschitz constants as time approaches zero.
> >    - Stabilizing regression or reallocating generative capacity offers one interpretation, but fundamentally, Diffusion Models learn the score within the integral of PF-ODE, while Consistency Models learn the entire integral. This does not suggest a fundamental difference between the literature or approaches, particularly in comparison to [2].
> >    - Additionally, the comment on [2] appears inaccurate; their results demonstrate that the best truncation time is not always "the smaller, the better" (see Fig. 8).

---

> ### Author Response · Authors · 2024-11-29
>
> ## The paper lacks ablation studies for certain hyperparameters (refer to W3).
>
> The hyperparameters the reviewer mentioned in W3 are $N_B$ (which is equivalent to $\rho$), $\Delta_t$, and $\Delta_{t’}$. As we discussed above, we already provide the ablation for $\rho$ in our paper (see Figure 4(c)). $\Delta_{t}$ represents “the nonnegative difference between two consecutive time steps $t$ and $s$” (i.e., $\Delta_{t}=t-s$), and it directly follows ECT’s definition (see Sec. 3.3 in [a]). $\Delta_{t’}$ is a special case of $\Delta_{t}$ when setting $t=t’$. Again, this is a general design choice for consistency models and not introduced by our method. Can the reviewer provide a reason for why including ablation studies for $\Delta_t$ would help improve this submission which is about truncated training?
>
> ## Guidance on determining when to terminate Stage 1 training is missing. For instance, what level of degradation occurs if Stage 1 training is imperfect? Does this significantly impact one-step or two-step sampling?
>
> This is a good question. As mentioned earlier, we train the Stage-1 model until its 1-step FID fully converges, because the Stage-1 model serves as a boundary condition during the truncated training stage. An imperfect Stage-1 model will propagate less accurate learning signals to the Stage-2 model.
>
>
> | Stage-1 FID | Stage-2 FID |
> |-------------|-------------|
> | 2.97 | 2.61 |
> | 2.82 | 2.54 |
>
> Table. Effects of Stage-1 FIDs on Stage-2 FIDs evaluated on CIFAR-10.
>
> Here, we see that the stage-1 model with FID=2.97 yields the final FID of 2.61. If we start with a better stage-1 model (FID=2.82), we get a better final FID of 2.54. We will consider adding this result to the appendix.
>
>
> ## Regarding [1, 2], I believe the fundamental issue is similar: addressing the challenge of large Lipschitz constants as time approaches zero. …does not suggest a fundamental difference … particularly in comparison to [2].
>
> We clarify that TCM is motivated from a fundamentally different perspective compared to [1,2]. To our best knowledge, truncation in [1,2] is addressing the training instability issue *near zero* (exploding norm of the score function [1] and the large Lipschitz constants in [2]). However, we haven’t observed training instability issue *near zero* when training consistency models. In fact, the consistency function (that maps any intermediate point to clean data at zero) approaches the identity function as $t \to 0$ and thus becomes much easier to learn when the time step $t$ is close to zero.
>
> >This does not suggest a fundamental difference between the literature or approaches, particularly in comparison to [2].
>
> What [2] observes is that the time derivative of the optimal $\epsilon_\theta$ in DDPM goes to $\infty$ near $t=0$. However, this is not the case for consistency models. Consider a simple case where data distribution is $\mathcal N(0,1)$. Then, the noised distribution $p_t = \mathcal N(0,1+t^2)$. In this case, the consistency function is $f(x,t) = x/(1+t^2)$, and $\partial f/\partial t = 0$ when $t \to 0$.
>
> We also further clarify the specific difference between TCM and [2] from the methodology perspective: [2] proposes to reduce the Lipschitz constants in the time interval $[0, t’)$ by sharing the timestep conditions in this interval. However, TCM first performs consistency training in the whole time interval (Stage 1) and then performs consistency training in the time interval $[t’, T]$ (Stage 2). As we see, when it comes to truncated training, [2] focuses on stabilizing the training in the early timesteps $t \in [0, t’)$ while TCM completely gets rid of training in the early timesteps $t \in [0, t’)$.
>
>
>
> ## Additionally, the comment on [2] appears inaccurate; their results demonstrate that the best truncation time is not always "the smaller, the better" (see Fig. 8).
>
> We apologize for causing the confusion. We did not intend to say “the smaller, the better”, instead, we meant that the *lower bound* of the truncated time interval in [2] is **close to zero**, which is mentioned in [2] multiple times: “diffusion models: they frequently exhibit the infinite Lipschitz near the zero point of timesteps.”, “our approach can effectively reduce the Lipschitz constants near t = 0 to zero”, “our approach substantially alleviates the large Lipschitz constants which occur in the intervals whose noise levels are small”, “E-TSDM can effectively mitigate the large Lipschitz constants near t = 0”. However, in TCM, $t’$ (the *lower bound* of our truncated time interval) should **not be close to zero**.
>
> ## References
>
> [a] https://arxiv.org/abs/2406.14548

---

> > ### Comment · Reviewer_jsgK · 2024-11-29
> > **Thank you for the replies.**
> >
> > I appreciate the authors' clarification. Regarding hyperparameter ablation, I am particularly curious about whether the number of splits in a batch, $N_b$, impacts performance. This is especially relevant for institutions unable to afford large batch size training—how would this parameter affects the results?
> >
> > That said, my other concerns have been addressed, and I have raised my score accordingly.

---

> > > ### Author Response · Authors · 2024-11-29
> > >
> > > >This is especially relevant for institutions unable to afford large batch size training
> > >
> > > That's a good point! Indeed, $N_B$ is introduced for such a case where the batch size is limited. Before providing a more detailed explanation regarding $N_B$, note that our default choice of $\rho=0.25$ (i.e., a 1/4 of a batch is used for computing boundary loss) works well with the text-to-image setting with a relatively small batch size of 128 (see general response).
> > >
> > >  In TCM training, we have two losses: the boundary loss and the consistency loss. Say we have 128 samples in a batch. We could, for example, use 64 samples for computing the boundary loss and another 64 samples for the consistency loss (i.e., $\rho=0.5$). Fig. 4(c) shows that this simple choice of $\rho=0.5$ yields competitive performance to our best setting on CIFAR-10. However, when the batch size is limited, we can be more intelligent in how we invest our compute to reduce the gradient variance of the total loss more efficiently. Intuitively, while the boundary loss is computed on a single point t=t', the consistency loss is computed on the entire truncated interval and thus may have a larger gradient variance than the boundary loss. Hence, we may choose a lower $\rho$ value to invest our compute more in optimizing the consistency loss rather than the boundary loss. In CIFAR-10, this is not a significant problem as it is a small dataset, but it may have a greater effect on a larger scale. We recommend practitioners start from our default choice of $\rho=0.25$, as it works well across CIFAR-10, ImageNet64, and even on the text-to-image.

---

> > > > ### Comment · Reviewer_jsgK · 2024-11-30
> > > > **Suggestion**
> > > >
> > > > I appreciate the authors further elaborate the selection of $N_b$ and batch size. I suggest the authors to conduct ablation study on it for a revised version to provide a more supportive evidence. With this, my major concerns are addressed and I recommend an acceptance.

---

### Official Review · Reviewer_ra9j · 2024-11-03

**Soundness:** 3
**Presentation:** 3
**Contribution:** 3
**Rating:** 6
**Confidence:** 3

**Summary:**

The paper divides the training of consistency models into two stages, where the first stage is the standard one, and the second stage only learns the consistency function in a truncated time region $[t',T]$. The training loss for the second stage is a mixture of the consistency loss and an extra boundary loss in $[t',t'+\Delta t]$. After careful tuning of the weighing functions and timestep schedules, the second stage model can achieve better 1-step and 2-step FID on CIFAR-10 and ImageNet 64x64 than standard consistency models.

**Strengths:**

- The motivation is the difficulty of learning the consistency function across the whole time zone, which is sound. The second stage model only needs to map noised data to clean data in a limited time interval, which is an easier task, and it is no surprise to bring better fitting for the ODE trajectory.
- The phenomenon that the consistency training gradually weakens the model's denoising capabilities at small $t$ is well illustrated in Figure 2.
- The method achieves better FID on standard image datasets than previous consistency models, especially in 1-step generation.
- Ablation studies for the dividing time and the number of stages are comprehensive.

**Weaknesses:**

- The overall idea and motivation are actually not new. ECT already points out the trade-off between the denoising capacity and the consistency capacity, suggesting the initialization of a consistency model with a diffusion model. The authors' work, in my opinion, is to use a two-stage method to replace the dedicated iteration-dependent training schedule in ECT.
- The 2-step performance improvement is relatively marginal compared to ECT. As suggested by ECT, it is better to use a 2-step generation with a small model instead of a 1-step generation with a large one. Therefore, the actual benefits provided by truncated training are limited, especially considering that it requires careful tuning of the weighing functions and timestep schedules.

**Questions:**

- Could the authors discuss the relation with PCM [1]?
- What are the total training iterations and time for the first and second stages, respectively? Is the first stage exactly the same as previous consistency models? If so, why the authors only initialize the truncated training stage with ECT for EDM2-XL?
- I think a limit case for the truncated training is to only learn the 1-step mapping $T\rightarrow 0$, i.e., limiting the truncated region to a single point $T$. The model can be directly trained on noise-data pairs. Can this method achieve better 1-step FID?


[1] Phased Consistency Model

---

> ### Author Response · Authors · 2024-11-24
>
> We appreciate the reviewer’s valuable comments.
>
> ## The overall idea and motivation are actually not new. ECT already points out the trade-off between the denoising capacity and the consistency capacity
>
> What ECT observes is that the optimal hyperparameters differ between 1-step and 2-step generation, while what we observe is that during the training of **the same model**, the trade-off between denoising and generation occurs after some training iteration (Fig. 2). The difference in motivation leads to the largely different solutions: ECT simply suggests choosing the hyperparameters optimized for 1 or 2-step generation, while we explicitly amplify this trade-off (Fig. 3) via truncated training to improve 1-step generation. The degree of improvement in FID coming from the truncated training cannot be achieved by simply manipulating the existing hyperparameters.
>
>
> ## The 2-step performance improvement is relatively marginal compared to ECT.
>
> Our method is mainly designed for 1-step generation where the improvement over ECT is more significant. Nevertheless, on ImageNet 64x64 with EDM2-S, our 2-step FID is 2.31, which surpasses the ECT's 2.79 by a nonnegligible margin (about a 17% reduction). The larger performance gain observed on the smaller model (EDM2-S) further validates the effectiveness of our truncated training procedure, by focusing the limited capacity on generation quality rather than denoising.
>
> In addition, we would like to point out that we contacted the ECT authors for more details in training ECT with EDM2-XL, but still could not reproduce its results, especially with a large batch size, which could be due to e.g. the training details for EDM2-XL that are not available to us. We believe that with the right configuration for large batch training, the 2-step FID of EDM2-XL could be improved further.
>
>
>
> ## As suggested by ECT, it is better to use a 2-step generation with a small model instead of a 1-step generation with a large one.
>
> We agree, but we think what is even better is to improve the one-step generation **without increasing the model size**. This is the main purpose of TCM. As shown in Table 2, our one-step FID outperforms all existing consistency model baselines and, with EDM2-S, even matches ECT's two-step FID (2.88 for TCM one-step FID vs 2.79 for ECT two-step FID). Also, TCM can also do multi-step sampling (by choosing t greater than the splitting time t’), so one can still target 2-step generation with a small model.
>
> ## Therefore, the actual benefits provided by truncated training are limited, especially considering that it requires careful tuning of the weighting functions and timestep schedules.
>
> We discuss the hyperparameters introduced by truncated training in the general response. We largely adopted the training hyperparameters from the ECT paper, with a few adjustments tailored specifically for our truncated training procedure. Our modifications to the ECT settings include employing a Student-t distribution for time sampling and tuning the coefficient for the boundary loss and the dividing time $t'$.
>
>
> We also provide a good guiding principle to determine their values, as discussed in Section 4.4. When developing our method, we intentionally avoided tuning these hyperparameters for any specific model or dataset and **used exactly the same values for all experiments to ensure the generalizability of our method.** We observe that these hyperparameters can easily be transferred from smaller to larger models/datasets, and their results are stable with a wide range of values. Hence, the additional complexity of our framework is marginal.
>
>
>
>
> ## Could the authors discuss the relation with PCM?
>
> We reviewed PCM and found that its idea is similar to multistep consistency models [1]. PCM and multistep consistency models are designed to improve multi-step generation by predicting the intermediate points on the PF ODE. In contrast, our model always predicts the data endpoint of the PF ODE as our goal is a one-step generation. Truncated training necessitates learning the boundary condition using our new parameterization and time sampling distribution, unlike these two papers. We will add a discussion of PCM alongside the multi-step consistency models in the related work section.

---

> ### Author Response · Authors · 2024-11-24
>
> ## What are the total training iterations and time for the first and second stages, respectively? Is the first stage exactly the same as previous consistency models?
> We thank the reviewer for pointing out the missing training details. The number of iterations is as follows:
> - CIFAR-10: Stage 1: 250K iterations, Stage 2: 200K iterations
> - ImageNet 64x64
>   - EDM2-S: Stage 1: 150K iterations, Stage 2: 120K iterations
>   - EDM2-XL: Stage 1: 100K iterations (we used the pre-trained checkpoint), Stage 2: 40K iterations
>
> The FID evolution on CIFAR-10 and ImageNet 64x64 (EDM2-S) during training is shown in Fig. 1(b). We have reflected this in the appendix.
>
> We did not try to exactly match the training iterations of previous works and used what worked best for our models. In all cases, stage 2 models outperform stage 1 models (see Fig. 1(b)).
>
> ## Why the authors only initialize the truncated training stage with ECT for EDM2-XL?
>
> That’s a good point. We tried hard to reproduce the results of ECT for EDM2-XL (the codebase for ImageNet was not released at the time of the submission). We contacted the ECT authors for more details in training ECT with EDM2-XL, but still could not reproduce its results, especially with a large batch size, which could be due to e.g. the training details for EDM2-XL that are not available to us. We thus ended up using the checkpoint provided by the authors.
>
> It is noteworthy that our truncated training procedure succeeds even when the standard, untruncated training (ECT with EDM2-XL) fails to converge. This observation suggests that the truncated training can be more stable. While stabilizing the original consistency model training is an important research direction, it falls outside the primary scope of our paper.
>
>
>
>
>
> ## I think a limit case for the truncated training is to only learn the 1-step mapping T -> 0, i.e., limiting the truncated region to a single point. The model can be directly trained on noise-data pairs. Can this method achieve better 1-step FID?
>
> > The model can be directly trained on noise-data pairs.
>
> Note that we do not use offline noise-data pairs in our training. That case corresponds to the knowledge distillation [2] (initialized with a pre-trained consistency model) and is not a consistency model anymore. Knowledge distillation methods not only necessitate the computationally expensive construction of offline noise-data pairs but also generally struggle to match the performance of state-of-the-art generative models, as evidenced in Table 1.
>
>
> Alternatively, within the TCM framework, we can explore shifting the dividing time $t'$ closer to $T$. However, as demonstrated in Table 3, increasing $t'$ beyond a certain threshold leads to performance degradation. Empirically, we observe that $t'=1$ yields the best results, better capturing the point in the probability flow ODE where the task transitions from denoising to generation.
>
>
>
>
>
>  ## References
>
> [1] Jonathan Heek, et al., Multistep Consistency Models, 2024.
>
> [2] Eric Luhman and Troy Luhman, Knowledge distillation in iterative generative models for improved
> sampling speed, 2021.

---

> > ### Comment · Reviewer_ra9j · 2024-11-28
> >
> > Thank the authors for the detailed response. I have raised my score.

---

### Official Review · Reviewer_uC7B · 2024-11-05

**Soundness:** 2
**Presentation:** 3
**Contribution:** 3
**Rating:** 8
**Confidence:** 5

**Summary:**

This work proposes a novel training technique for consistency models by truncating the time intervals into two splits: one is the boundary/denoising part near $t=0$, and the other is the consistency/generation part near $t=T$. By training with a weighted sum of the two objectives and introducing a dedicated proposal distribution for sampling the time, the proposed method beats previous 1-step and 2-step models on CIFAR10 and ImageNet64.

**Strengths:**

- The paper is well-written and is easy to follow. The math derivations are technically correct.
- The idea of truncating is novel, and the proposed method can further address the training instability of consistency models which are important to the community.
- The empirical results are strong, showing the effectiveness of the method.

**Weaknesses:**

It is unclear whether the effectiveness is from the truncation or is just from the changing of proposal distribution by focusing more on the boundary parts. Specifically, the major techniques in this paper include two parts:

1. two-stage truncated training to avoid the overtraining of denoising tasks;
2. changing proposal distribution to focus more on the boundary conditions.

However, with only the first method, the training still diverges (L241-L242), which shows that the part 2 seems to be more important. Intuitively, as the supervision signals of consistency training only come from the boundary condition, the model will suffer from error accumulations as t increases from 0 to T. Therefore, focusing more on the boundary parts seem to be a natural idea to address such error accumulations and may be the reason why the proposed method is more stable than previous methods. Given such understanding, it is unclear whether the first part (truncation) is even necessary.

Therefore, this paper does not fully sound to me. It would be great if the author could conduct more rigorous ablations for the part 2, such as:

- Training consistency models by the original loss (i.e. the method used in stage-1) with the improved proposal distribution by focusing more on [0, t']. The optimal proposal distribution may be different from the stage-2 so it should be tuned a little bit for such task. Will the optimal result be comparable to the 2-stage training method?

Besides, there is another minor weakness but I think it does not affect the conclusions in this paper:

- Consistency models include both consistency training (CT) and consistency distillation (CD), but this paper only shows the results of CT. Note that the original CT has the same gradient limit as CD as $\Delta t \to 0$. However, with truncation training, this conclusion will not hold anymore for t near to t'. What is the performance of CD by the truncated method?

**Questions:**

- Training consistency models by the original loss (i.e. the method used in stage-1) with the improved proposal distribution by focusing more on [0, t']. The optimal proposal distribution may be different from the stage-2 so it should be tuned a little bit for such task. Will the optimal result be comparable to the 2-stage training method?
- Consistency models include both consistency training (CT) and consistency distillation (CD), but this paper only shows the results of CT. Note that the original CT has the same gradient limit as CD as $\Delta t \to 0$. However, with truncation training, this conclusion will not hold anymore for t near to t'. What is the performance of CD by the truncated method?

(Please refer to Weakness section for detailed explanations)

---

> ### Author Response · Authors · 2024-11-24
>
> We appreciate the reviewer’s valuable comments.
>
> ## However, with only the first method, the training still diverges (L241-L242), which shows that the part 2 seems to be more important. It is unclear whether the first part (truncation) is even necessary.
>
> We clarify that the truncated training and the proposed time sampling distribution are not two independent design choices but rather one cannot be separated from the other. Without the proposed time sampling distribution, as $\Delta_t$ goes to zero, the truncated training permits a trivial solution (e.g. a constant function). In preliminary trials, we also observed empirically that this design choice does not work well.
>
> ##  It would be great if the author could conduct more rigorous ablations: Training consistency models by the original loss (i.e. the method used in stage-1) with the improved proposal distribution by focusing more on [0, t'].
>
> That’s a good point! Such a strategy is already discussed in ref. [1], with a similar reasoning to the reviewer: "errors from minimizing consistency losses in smaller noise levels can influence larger ones and therefore should be weighted more heavily" (note that the time sampling distribution and the weighting functions are interchangeable). As such, our current weighting functions are already designed to emphasize the losses at smaller $t$'s following prior work. However, overemphasis on smaller $t$'s even further does **the exact opposite of what we want (as it increases the capacity dedicated for denoising).**
>
> Nevertheless, we run experiments to see the effect of changing the weighting function to focus more on smaller $t$'s. On CIFAR-10, from the default weighting function $1/\Delta t$, we simply introduce an additional power variable $p$ to the previous weighting function, i.e.,
> $$\frac{1}{\Delta_{t,p}} = \frac{1}{(1+8 \cdot \text{sigmoid}(-t))(1-r) t^p}$$
> , to adjust the emphasis on smaller $t$'s. See [link](https://anonymous.4open.science/r/ICLR2025-rebuttal-2C2B/weightings.png) for visual comparisons. The results are shown below:
>
> | | p=1 (default) | p=1.5 | p=2 |
> | --- | --- | --- | --- |
> | FID | 2.82         | 3.48     | 27.57   |
>
> These results suggest that the improvement from our method is not merely due to the increased emphasis on smaller t's but rather the explicit redistribution of the capacity.
>
>
> ## CT has the same gradient limit as CD. However, with truncation training, this conclusion will not hold anymore for t near to t'. What is the performance of CD by the truncated method?
>
> Truncated consistency models are also consistency models with a specific parameterization and time sampling distribution. Therefore, the equivalence between CT and CD in the limit still holds for TCM (Eq. 35 in Appendix does not assume any specific parameterization of $f$). That is, the relationship between CT and CD in TCM should be the same as that in the original consistency models (the former comes with an unbiased but high-variance gradient, while the latter has a biased but low-variance gradient). For the performance of CD in TCM, please see the general response where we train TCM with CD on the text-to-image setting.
>
>
> ## References
> [1] Yang Song, Prafulla Dhariwal, Improved Techniques for Training Consistency Models, 2023.

---

> > ### Comment · Reviewer_uC7B · 2024-11-26
> > **Thank you**
> >
> > Thank you for the response which addressed my concerns. I'd like to recommend this paper to be accepted so I raised my score to 8.

---

### Author Response · Authors · 2024-11-24
**General response**

We appreciate the reviewer’s valuable comments. Here, we provide experimental results and also discuss the points raised by several reviewers.

## Text-to-image results
Upon the reviewers' request, we train our TCM on the text-to-image setting, using consistency distillation with a fixed classifier-guidance scale of 6.

| | Stage 1 | Stage 2 |
|:---:|:---:|:---:|
|FID↓| 20.50 | **17.33** |

Table. Zero-shot FID scores on MSCOCO dataset measured with 30K generated samples.

The table above shows the effectiveness of TCM in the text-to-image setting. Note that we use the same hyperparameters for truncated training as other experiments in our paper. See Appendix A for a visual comparison and more details.

## Q: Truncated training introduces additional hyperparameters--how do we justify this?
Our method does introduce a few additional hyperparameters, but we believe this is not always bad as long as one of these conditions is met: 1) there is a good guiding principle to determine their value, 2) the hyperparameters can easily be transferred from the smaller to larger models/datasets, or 3) the results are stable with a wide range of values.

The following are the hyperparameters introduced by truncated training:
- Dividing time $t'$: Simply setting $t'$ to 1 works well across different datasets (from CIFAR-10 to ImageNet 64x64) and different model sizes (from 60M to 1.1B parameters).
- Boundary loss weight $w_b$ and boundary ratio $\rho$: these are stable with a wide range of (from 0.1 to 0.5 for $\rho$ and from 0.1 to 1 for $w_b$), as shown in Fig. 4(c).
- Student's t-distribution parameter $\mu,\sigma,\nu$: rather than the specific value of these, what matters is the shape of the distribution--it should put a large mass around $t'$ while still putting some mass on the tails, as discussed in Sec. 4.4. The values of these parameters can easily be chosen following this guideline.


Note that for $\Delta_t$, we directly follow ECT without modification as it is not particularly related to the idea of truncated training.

Importantly, when developing our method, **we intentionally avoided tuning these hyperparameters** for any specific model or dataset and **used exactly the same values for all experiments** to ensure the generality of our method. We hope this will address the reviewers' concerns about the complexity of our method.

---

### Meta-Review · Area_Chair_od9S · 2024-12-17

**Metareview:**

This paper proposes a method to improve the training of consistency models by addressing the capacity challenge of simultaneously learning denoising and generation. It introduces a two-stage approach: first training a standard consistency model for denoising, then training a separate generation model. A key challenge is designing the loss and weighting function to enforce boundary conditions between time ranges. The proposed method achieves stable training with progressively decreasing FID and outperforms existing work.

All reviewers find the paper well-written, interesting, and with strong empirical results.

**Additional Comments On Reviewer Discussion:**

All reviewers remain positive about the paper during the rebuttal.

---

### Decision · Program_Chairs · 2025-01-22

Accept (Poster)